# Transient inhibition of 53BP1 increases the frequency of targeted integration in human hematopoietic stem and progenitor cells

Ron Baik [1,2,3], M. Kyle Cromer [1,2], Steve E. Glenn[4], Christopher A. Vakulskas [4], Kay O. Chmielewski[5,6,7], Amanda M. Dudek[1,2], William N. Feist [1,2], Julia Klermund [5,6], Suzette Shipp[1,2], Toni Cathomen [5,6], Daniel P. Dever[1,2] & Matthew H. Porteus[1,2] ✉

Genome editing by homology directed repair (HDR) is leveraged to precisely modify the genome of therapeutically relevant hematopoietic stem and progenitor cells (HSPCs). Here, we present a new approach to increasing the frequency of HDR in human HSPCs by the delivery of an inhibitor of 53BP1 (named "i53") as a recombinant peptide. We show that the use of i53 peptide effectively increases the frequency of HDR-mediated genome editing at a variety of therapeutically relevant loci in HSPCs as well as other primary human cell types. We show that incorporating the use of i53 recombinant protein allows high frequencies of HDR while lowering the amounts of AAV6 needed by 8-fold. HDR edited HSPCs were capable of long-term and bi-lineage hematopoietic reconstitution in NSG mice, suggesting that i53 recombinant protein might be safely integrated into the standard CRISPR/AAV6-mediated genome editing protocol to gain greater numbers of edited cells for transplantation of clinically meaningful cell populations.

The CRISPR/Cas9 technology has democratized genome editing and is accelerating the development of a new class of genetic medicines in which the genome of cells is modified with single nucleotide precision[1]. Following a targeted break on both strands of DNA (a double-strand break (DSB)), repair typically occurs through either homologous recombination (HR), non-homologous end-joining (NHEJ), or microhomology-mediated end-joining (MMEJ) pathways. NHEJ and MMEJ are used in all stages of the cell cycle to repair spontaneous breaks and can result in insertions or deletions (indels) of one to several bases at the site of the break. In HR mediated repair, the recombination machinery uses an undamaged donor, usually the sister chromatid, as a template for the recombination process. In genome editing by homology directed repair (HDR), an exogenous donor DNA

molecule replaces the sister chromatid as a template for repair of the induced DSB. When the donor is provided as a DNA molecule with long homology arms (>50 basepairs, but ideally >400 basepairs) the RAD51 dependent natural HR pathway is used. This gene targeting donor can be delivered to cells using a non-integrating adeno-associated virus (AAV). When the donor is provided as single-stranded DNA oligonucleotide (ssODN), a RAD51 independent single-stranded template repair (SSTR) process is used[1,2]. Leveraging these endogenous DNA repair processes, genome editing can be used to generate precise genomic modifications in living cells for research and therapeutic purposes.

Hematopoietic stem and progenitor cells (HSPCs) have the ability to repopulate an entire hematopoietic system for the lifetime of the

[1]Institute for Stem Cell Biology and Regenerative Medicine, Stanford University School of Medicine, Lorry I. Lokey Stem Cell Research Building, 265 Campus Drive, Stanford, CA, USA. [2]Department of Pediatrics, Stanford University School of Medicine, Stanford, CA, USA. [3]Molecular Biology Program, Sloan Kettering Institute, Memorial Sloan Kettering Cancer Center, New York, NY, USA. [4]Integrated DNA Technologies, Inc., Coralville, IA, USA. [5]Institute for Transfusion Medicine and Gene Therapy, Medical Center – University of Freiburg, 79106 Freiburg, Germany. [6]Center for Chronic Immunodeficiency, University of Freiburg, 79106 Freiburg, Germany. [7]Ph.D. Program, Faculty of Biology, University of Freiburg, 79104 Freiburg, Germany. ✉e-mail: mporteus@stanford.edu

person. Transplantation of HSPCs is standard of care for both malignant and non-malignant diseases. Genome editing of HSPCs has the potential, therefore, to provide durable treatments for the lifetime of the patient based on the natural biologic properties of HSPCs. While genome editing-based therapies for hematological diseases such as sickle cell disease and β-thalassemia have entered clinical trials using INDEL based approaches, only a single trial using HR to directly correct an underlying pathologic disease causing variant has entered clinical trials by 2023[3]. Preclinical studies involving HDR-mediated gene editing highlight that the frequencies of HDR can be remarkably high, with frequencies ranging from 30-70% in primary human hematopoietic stem and progenitor cells (HSPCs), T cells, induced pluripotent stem cells (iPSCs), basal cells, and mesenchymal stromal cells (MSCs)[4–11]. We have shown that recombinant adeno-associated viral vectors of serotype 6 (AAV6) can efficiently deliver DNA to the nucleus of HSPCs where it can be used as a template for HDR, allowing the introduction of custom genome edits at the site of the CRISPR-mediated DSB[5–7,9]. However, current xenograft studies have shown that editing frequencies in the pre-transplantation population of HSPCs are consistently higher than editing frequencies in the population of cells that successfully engraft in mice in vivo[5,12–14]. Some possible explanations for this phenomenon are that, among the heterogeneous population of HSPCs, (1) long-term repopulating hematopoietic stem cells (LT-HSCs) are more refractory to the editing process, (2) the editing process negatively impacts LT-HSC repopulation and differentiation capacity, and/or (3) LT-HSCs are more prone to apoptosis following donor DNA transduction[15,16]. Both the degree of edited-cell chimerism in the bone marrow as well as the total number of HDR edited cells engrafted may be important factors for achieving clinically efficacy and thus are important subjects for study, especially for diseases where there is not a disease-specific cellular selective advantage.

One of the key questions in the metabolism of DSBs is how a cell chooses to repair the break. A key step is the mechanism by which the broken strand is processed. The NHEJ pathway is activated by free DNA ends leading to the recruitment of proteins such as Ku70/Ku80 dimer to bind the DNA end and marking of the site of the break by phosphorylation of H2AX and recruitment of proteins such as 53BP1 to a megabase domain surrounding the break[17]. In contrast, the HDR pathway is initiated by end resection to generate 3′ single-strand tails, a process facilitated by the protein CtIP[18]. One of the mechanisms by which 53BP1 biases repair away from HDR and towards NHEJ is by inhibiting binding of BRCA1, a protein required for homologous recombination[19]. Previously, Canny et al. discovered that inhibition of 53BP1 through an engineered ubiquitin variant called "i53", delivered by either plasmid transfection or AAV, increased the frequency of Cas9-mediated HDR in human cancer cell lines[20]. Additionally, Nambiar et al. discovered that an engineered RAD18 ("e18") variant that suppresses 53BP1 can also be used to stimulate HDR[21]. These studies demonstrate that i53 or e18 inhibit accumulation of 53BP1 at DSBs, thereby facilitating repair of the break by HR rather than NHEJ.

Though these results were quite promising, the two methods of delivery that were tested are likely to be problematic for delivery into cells such as HSPCs. First, transfection of naked DNA into primary human cells results in the induction of a toxic type I interferon response that negatively impacts viability and editing frequency[22,23]. Second, while AAV has been shown to be relatively well-tolerated by primary cells[24], the slow kinetics of expression from AAV transduction would mean that i53 protein would not inhibit 53BP1 when the Cas9-induced break was made, especially if the Cas9 nuclease is delivered as ribonucleoprotein complex. There is also the risk of random integration of the vector causing persistent i53 expression which would exacerbate any genotoxic effects[25,26].

As an extension to the discovery of the inhibition of 53BP1 reported by Canny et al.[20], De Ravin et al. showed that transient inhibition of 53BP1 using i53 mRNA significantly increases HDR and can

achieve highly efficient functional correction of engrafting HSPCs[27]. In line with the recent reports as well as our experience that directly delivering recombinant proteins to primary cells is an effective approach to genome editing, we hypothesized that direct delivery of i53 in the form of protein would result in a non-toxic, kinetically active, yet transient and effective means of increasing HDR in human HSPCs and other cell types. In this work, we discovered that delivery of i53 recombinant protein effectively increases the frequency of HDR-mediated genome editing at a variety of therapeutically relevant loci in HSPCs. Delivery of i53 protein by electroporation leads to inhibition of 53BP1 that is transient, and this strategy results in improvement in the frequencies of edited HSPCs that successfully engraft in mice in vivo. 53BP1 is a key player in the NHEJ pathway and altering its function could potentially increase the risk of genome rearrangements. Therefore, it is essential to evaluate potential risks and understand the mechanisms that underlie genome instability. Several important genome instability risks in gene editing include: (1) off-target effects, where Cas9 enzyme can unintentionally introduce DSBs at sites that resemble the target sequence, (2) unintended insertions, deletions and rearrangements can result from the error-prone nature of NHEJ repair pathway, and finally (3) unwanted rearrangements and insertions can result from HDR pathway, which might not always be precise.

Therefore, we hypothesize that inhibiting 53BP1 through peptide delivery, rather than mRNA delivery, shortens the duration of inhibition of natural DNA repair. This eliminates the need for prolonged modulation of the natural DNA repair process and mitigates the risk of triggering a Type I interferon response that can be induced in HSPCs even with modified mRNAs[24]. We show that we can use i53 protein to reduce the AAV6 MOI needed to achieve the same frequencies of HDR, accompanied by a reduction in the p53 response that is triggered by AAV transduction and increase the colony-forming units from gene targeted human HSPCs[15]. Therefore, delivery of i53 protein may be a viable strategy to increase the frequency of HDR in HSCs in the clinic and may be applicable for a broader use in a variety of therapeutically relevant human primary cell types including MSCs and airway basal stem cells.

## Results

### i53 peptide rescues AAV transduction toxicity in CD34[+] HSPCs

It has been shown that HDR frequencies can be increased by inhibiting 53BP1 via plasmid transfection of the inhibitory peptide in cancer cell lines[20]. The rationale is that inhibition of 53BP1 would bias cells to repair using the HDR pathway. Plasmid transfection, however, is not tolerated in primary human cells because of the activation of the Type I interferon response from naked cytosolic DNA sensing[22,23]. In addition, the prolonged inhibition of 53BP1 might cause genomic instability that would counter-balance any potential increase in HDR frequency. It has been shown, for example, that mice with knockouts of 53BP1 are susceptible to lymphomas derived from the hematopoietic system[28,29]. Therefore, we tested whether transient co-delivery of the i53 molecule as a purified peptide along with Cas9-RNP could increase HDR in primary human HSPCs.

We have shown in previous studies that gene-specific AAV6 donor MOIs in the range of 2500–10,000 can be used to achieve high levels of HDR in HSPCs[4–6,24]. However, there are indications that high AAV6 MOIs may impair cell fitness, affecting both cell viability and stem cell function[5,15]. Therefore, we determined whether increasing AAV6 MOIs indeed does impair HSPC function by performing colony-forming unit (CFU) assays using HSPCs transduced with AAV6 with MOIs ranging from 625 to 5000. To assess the effect of AAV6 transduction on HSPC function in isolation (in the absence of DSB induced by RNP delivery), HSPCs resuspend in P3 buffer (Lonza) were electroporated (using the Lonza 4D Nucleofector program DZ-100) and transduced only with a therapeutic AAV6 donor template which includes a homologous sequence (consisting total homology sequence of 960 bp) to *HBB*

designed to revert the SCD-causing Glu6Val mutation to wild-type alleles, while also introducing silent mutations to prevent Cas9 re-cutting and premature cross over (Supplementary Fig. 1b and Supplementary Tables 1 and 2)[5]. The transduced HSPCs were then plated in semisolid methylcellulose media, which supports the growth of multiple progenitor cells (myeloid: CFU-GM; erythroid: BFU-E and CFU-E; and mixed myeloid and erythroid: CFU-GEMM). Fourteen days after plating, colonies were scored and we found a dose-dependent decrease in colony-forming ability of HSPCs as AAV6 vector genomes increased (Fig. 1a). To ascertain the mechanism behind decreased colony formation with higher MOIs of AAV6, we investigated the activation of p21 which has been observed in HSPCs following AAV transduction[15,30]. p21 is an important effector protein and is a direct transcriptional target following p53 activation. p21 inhibits the cyclin kinases that are necessary for cell cycle progression. Upon DNA damage (i.e., DSBs) or exposure to single-stranded DNA as in AAV vectors, expression, and activation of p21 triggers G1 cell cycle arrest, which allows the damaged cells to repair the DNA breaks. However, prolonged activation of p21 can lead to a chronic state of senescence or apoptosis[31]. Therefore, we quantified p21 mRNA levels using droplet digital PCR (ddPCR) 24 h post-AAV6 transduction of HSPCs and found that AAV6 MOI directly correlates with p21 expression (see Methods) (Fig. 1b).

To counteract this strong p53 response during genome editing, we investigated ways of reducing AAV6 MOIs while still achieving high frequencies of HDR. To test our hypothesis that transient delivery of i53 peptide may improve HDR in CD34+ HSPCs, we determined editing frequencies using an AAV6 MOI of 625 across a range of i53 concentrations. *HBB*-edited HSPCs were harvested 2 days post-electroporation, together with mock control (electroporation only), and then analyzed for modification frequencies of the *HBB* alleles using nested droplet digital PCR (ddPCR) as previously described[32] (see Methods; Supplementary Fig. 1b and Supplementary Tables 3 and 4). We found that i53 peptide increased HDR in HSPCs in a dose-dependent manner by 9%, 28%, 37% and 43% at 375, 750, 1500 and 2250 µg/mL, respectively (Fig. 1c). Importantly, i53 peptide concentrations up to 5000 µg/ml did not result in notable decrease in the viability of HSPCs (Supplementary Fig. 1a), and dosage of i53 peptide beyond 1500 µg/mL did not result in any additional increase in HDR (Fig. 1c). Based on these results, we used 1500 µg/mL of peptide in subsequent experiments unless noted otherwise. As an additional measurement of DSB repair, we investigated the capacity of i53 peptide to bias NHEJ towards HDR in HSPCs by simultaneously measuring both INDELs and HDR. We found that indels were correspondingly reduced in cells edited in the context of increasing concentrations of i53 (Fig. 1c). We determined whether the incorporation of i53 peptide into our editing protocol would allow reduction in AAV MOI while still achieving high frequencies of HDR. To do so, we determined frequencies of HDR in edited HSPCs across a range of AAV6 MOIs (625–5000) in cells treated with or without i53 peptide and found that we could significantly enhance HDR while reducing the amount of AAV6 donors by 8-fold (67.1% HDR by 625 MOI vs. 64.9% HDR by 5,000 MOI) (Fig. 1d).

### Effect of i53 peptide on colony-forming unit activity and p53 pathway activation

We determined if i53 peptide negatively affects progenitor cell viability and hematopoietic differentiation capacity by colony-forming unit (CFU) assays. We performed CFU assays using HSPCs edited and transduced with AAV6 with MOI at either 625 or 2500 that either have or have not been treated with i53 peptide. Following electroporation with RNP and with or without i53 peptide, HSPCs were transduced with AAV6 and plated in methylcellulose media. Fourteen days after plating colonies were assessed and we found that the inclusion of i53 had no discernible effect on colony-forming ability of HSPCs (Fig. 2a). However, we found that there is an AAV6 MOI-dependent decrease in

colony formation of edited HSPCs – 61% of HSPCs transduced with 2500 MOI of AAV6 form colonies, 68% of HSPCs with 2500 MOI + i53 peptide, 89% with 625 MOI, and 92% with 625 MOI + i53 peptide (Fig. 2a). In additional experiments, we found that cells edited with MOI of 625 + i53 peptide resulted in improvement of both HDR frequencies and colony-forming abilities in comparison to cells edited with high MOI of 5,000 and no i53 peptide treatment (Fig. 2b). These data indicate that the inclusion of i53 peptide allows us to achieve high frequencies of HDR with lower AAV MOIs, the reduction of which maintains HSPC viability and differentiation capacity (Fig. 2a, b and Supplementary Fig. 1a)[9].

To shed light on the role that p21 activation may be playing in this phenomenon, we used ddPCR to quantify p21 expression levels across a range of MOIs at various timepoints post-editing. Consistent with prior data, we observed an AAV6 dose-dependent effect on p21 expression (Fig. 2c). We also observed that inclusion of i53 into the editing process reduced p21 activation at both high and low AAV6 MOIs (Fig. 2d) without reducing HDR frequencies (Fig. 2e). These findings corroborate that the use of AAV6 during gene editing can trigger a strong p53-related transcriptional response, while the addition of i53 peptide can achieve high frequencies of HDR at lower AAV6 MOIs while also minimizing p21 upregulation (Fig. 2e).

### i53 recombinant protein improves HDR in human primary cells at multiple loci

We tested gene targeting at four different loci (*HBB, CCR5, IL2RG and HBA1*)[4,5,33–35] in CD34+ HSPCs from either cord blood or plerixafor-mobilized healthy donor cells (Supplementary Table 1). The cells were electroporated with Cas9-RNP using previously described chemically modified sgRNAs[36] either with or without i53 peptide and transduced with AAV6 donor vectors at an MOI of 5000[6]. At all four clinically relevant loci, we observed an increase in HDR frequencies (ranging between 29% and 42% increase when i53 peptide was co-delivered) (Fig. 3a and Supplementary Fig. 2a, b). As an additional measure of DSB repair, we investigated the capacity of i53 peptide to decrease the frequency of INDELs in the corresponding HSPCs. We found that the indels were decreased in the targeted cells with the use of i53 peptide (69% decrease for *HBB*, 21% decrease for *CCR5*, and 15% decrease for *IL2RG*) (Fig. 3b).

With the higher frequencies of HDR caused by including i53, we determined whether i53 might also increase the frequency of biallelic HDR within a given cell. To measure biallelic HDR, we targeted the *HBB* gene as previously[5] and transduced HSPCs with AAV6 donors to deliver GFP and mCherry integration cassettes at the corresponding Cas9 cutting site. Three days post-editing, we found that incorporation of i53 peptide increased frequency of both GFP and mCherry alone, as well as frequency of double positive cells (Fig. 3c). Biallelic targeting increased from 13.3% to 18.8% with the inclusion of i53 (an increase of ~41%).

We explored whether inclusion of i53 peptide during gene editing would increase gene targeting in additional clinically relevant primary human cell types (Fig. 3d). Basal stem cells (CFTR), T cells (TRAC) and MSCs (HBB) were edited with previously developed gene-editing reagents (Supplementary Table 1) designed for usage in their corresponding cell types[11,37,38] and we found that the incorporation of i53 peptide increased HDR frequencies by ~120% in basal cells, ~7% in T cells, and ~30% in MSCs (Fig. 3d).

### Transient effect of I53 peptide in CD34+ HSPCs following electroporation

When DNA repair pathways malfunction, translocations and other genome rearrangements might result, diminishing cell viability and increasing the chance of tumorigenic events[28,29]. Transient inhibition of 53BP1 through i53 peptide should, however, minimize these potential effects compared to sustained expression. To determine the

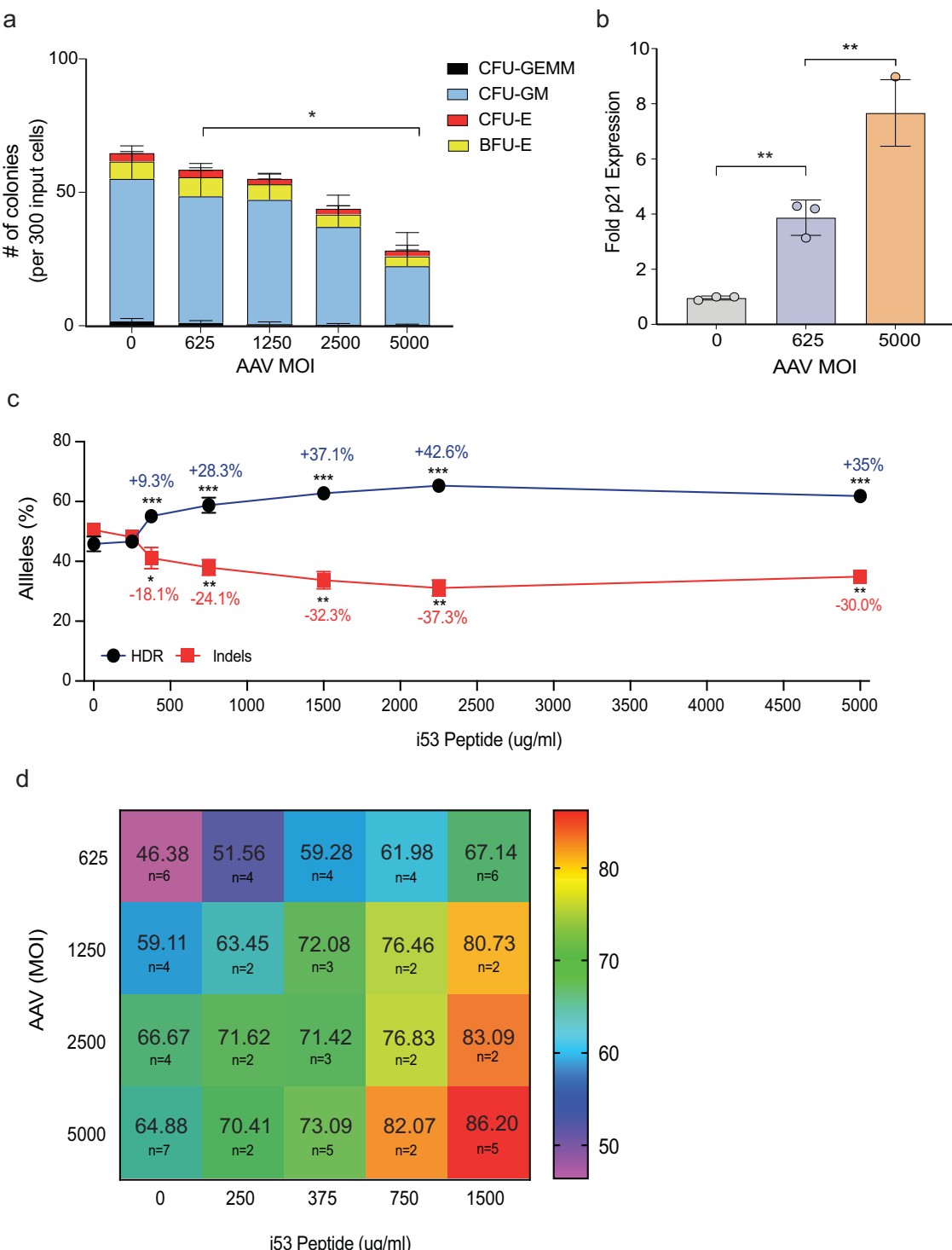

**Fig. 1 | Optimizing i53 peptide for targeting *HBB* locus in CD34⁺ HSPCs. a** HSPCs derived from umbilical cord blood were transduced with different MOIs of AAV6 were plated on methylcellulose and scored as CFU-E, BFU-E, CFU-GM, or CFU-GEMM based on morphology 14 days after plating. Data from $n = 3$ biological cord blood donors with technical replicates per donor and the mean ± SD depicted. $*P < 0.05$ ($P = 0.0144$) by two-tailed, unpaired $t$-test (the sum of all colonies formed per experiment were used to calculate the $P$ value). **b** p21 mRNA levels were assessed using droplet digital PCR (ddPCR) 24 h post-AAV6 transduction of HSPCs. Data from $n = 3$ independent biological replicates with mean ± SD depicted. $**P < 0.05$ ($P = 0.0014$ for 625 MOI vs Mock; $P = 0.0086$ for 5000 MOI vs 625) by two-tailed unpaired $t$-test. **c** CD34⁺ HSPCs were electroporated with Cas9-RNP and AAV6 with increasing amounts of i53 peptide, targeting at *HBB* locus with integration of Glu6Val AAV6 donor. HDR and indel frequencies were assessed by ddPCR. Data from $n = 2$ independent biological replicates performed with technical duplicates per biological donor. Mean ± SD depicted. $*P = 0.0147$; $**P < 0.01$ ($P = 0.0027$ for 750 vs 0; $P = 0.0021$ for 1500 vs 0; $P = 0.0012$ for 2250 vs 0; $P = 0.0020$ for 5000 vs 0) and $***P < 0.001$ ($P = 0.0005$ and $P = 0.0007$) by two-tailed unpaired $t$-test. All statistical tests were run in comparison to CD34⁺ HSPCs electroporated with Cas9-RNP and AAV6 with no i53 addition. **d** Heatmap illustrating HDR frequencies in response to various doses of AAV6 and i53 peptide. CD34⁺ HSPCs were electroporated with Cas9-RNP, 625 MOI to 5000 MOIs of AAV6 and 0–1500 ug/ml of i53 peptide. HDR-mediated outcomes were assessed by ddPCR data ranging from $n = 2$ to $n = 6$ independent biological replicates (precise $n$ values reported on the figure) with mean values depicted. **a–d** Source data are provided as a Source Data file.

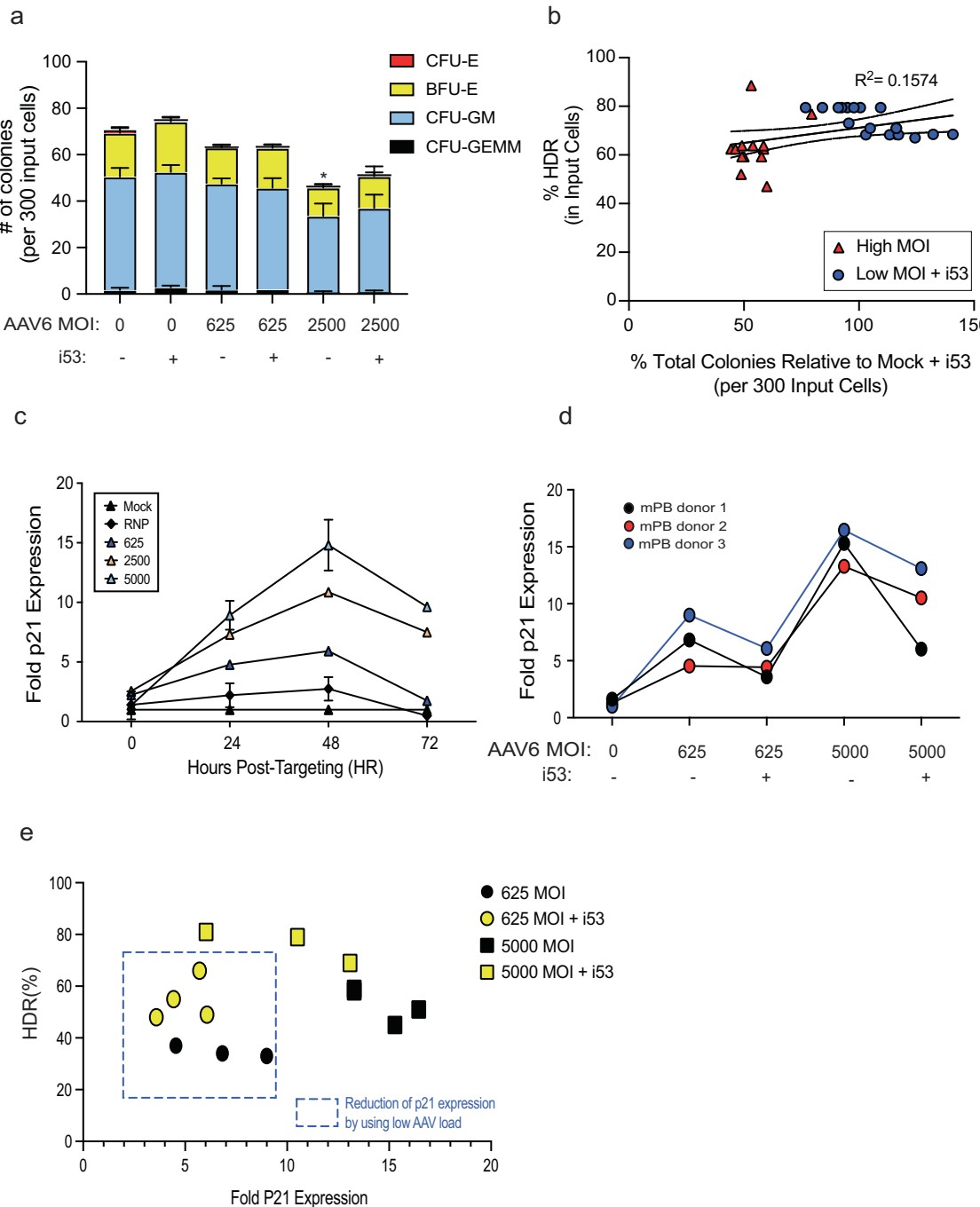

**Fig. 2 | i53 peptide rescues AAV transduction toxicity in CD34+ HSPCs. a** Edited HSPCs were plated on methylcellulose and scored as CFU-E, BFU-E, CFU-GM, or CFU-GEMM based on morphology 14 days after plating. Data from $n = 6$ independent biological replicates for mock, mock + i53, 625, 625 + i53 treated cells. Data from $n = 12$ independent biological replicates for 2500 and 2500 + i53 treated cells. Mean ± SD depicted. *$P < 0.05$ ($P = 0.0435$) by two-tailed unpaired $t$-test. **b** X-Y linear correlation between HDR frequency and % total colonies formed relative to control (mock electroporated and i53 peptide-treated HSPCs) on methylcellulose. Cells transduced with either high (5000) MOI of AAV, or low (625) MOI of AAV and treatment with i53 peptide are depicted. For negative control, HSPCs were mock electroporated and treated with i53 peptide prior to plating on methylcellulose. $P = 0.02$ by simple linear regression. **c** Kinetics of p21 expression assessed by ddPCR post-editing for up to 72 h. Data from $n = 3$ biological replicates with mean ± SD depicted. **d** Expression of p21 assessed by ddPCR 48 h post-editing. Data from $n = 3$ biological donors plotted. **e** X-Y linear correlation between HDR frequency and p21 expression in HSPCs. Reduction of p21 expression by using low MOI of AAV are highlighted in square brackets []. Data from $n = 3$ biological donors plotted. Data from $n = 4$ biological donors plotted for 625 MOI + i53 treated cells. **a**–**e** Source data are provided as a Source Data file.

duration of inhibition of 53BP1, we assessed the duration of the i53 peptide expressed in cells post-editing. Following the electroporation of HSPCs with RNP + AAV6 and i53 peptide, we collected cells at 1-h intervals from 0 to 4 h and at 24 and 48 h. The i53 peptide is tagged at its N-terminus with a His-tag enabling us to visualize the full-length protein via Western blot. We found that the peptide is rapidly degraded in the cells within 4 h of electroporation, highlighting that the protein is present in cells for only a short time (Fig. 4a). We assessed the impact of i53 on the kinetics of both HDR and indels following editing. CD34+ HSPCs were edited with Cas9-RNP and AAV6 donor with

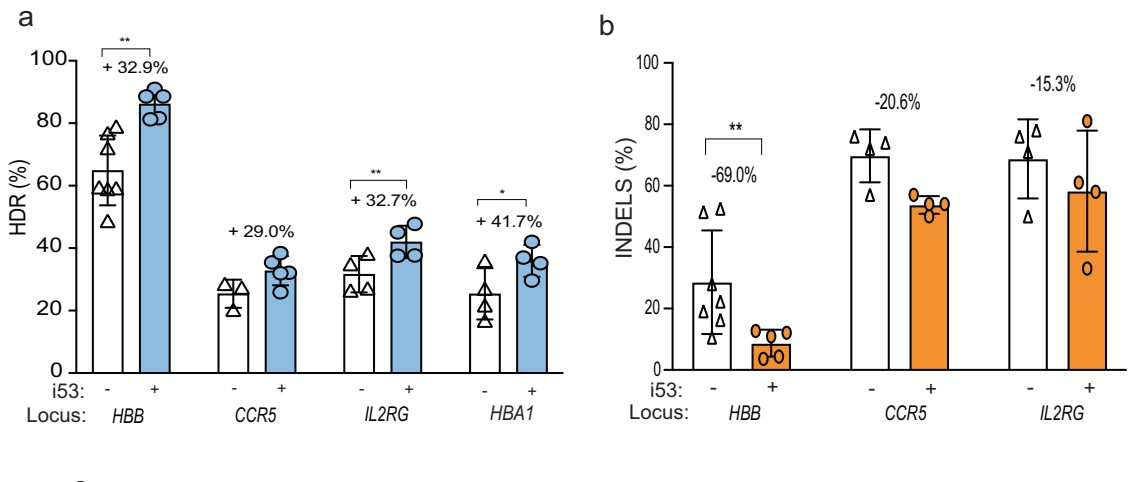

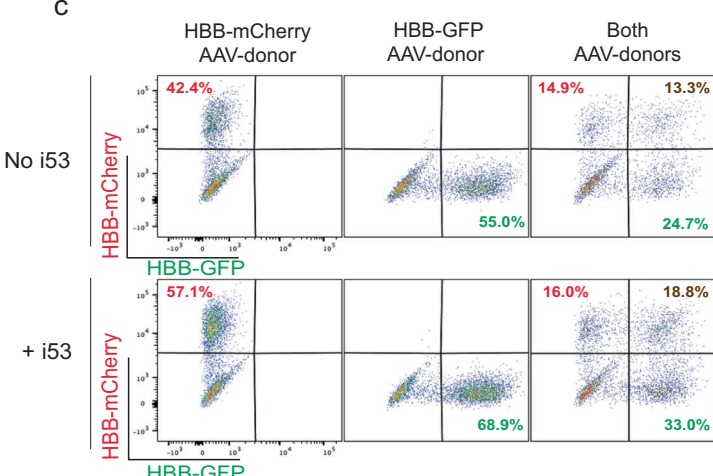

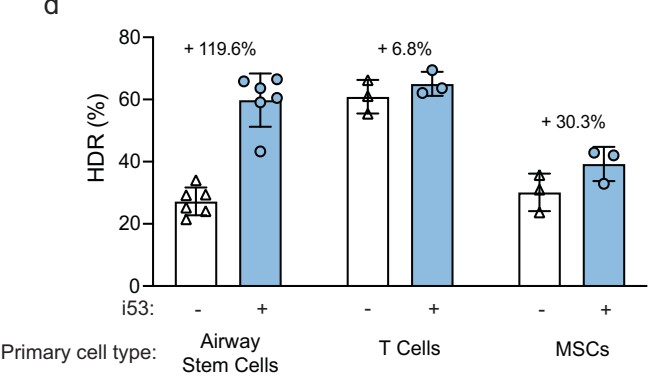

**Fig. 3 | Cas9-RNP and AAV6-mediated targeting of human primary stem cells using i53 peptide. a** CD34⁺ HSPCs were electroporated with Cas9-RNP and AAV6 with or without i53 peptide targeting different genes (*HBB, CCR5, IL2RG, HBA1*). HDR-mediated outcomes were assessed by ddPCR 2 days post-electroporation. Data from *n* = 7 independent biological replicates for *HBB* targeting without i53, *n* = 5 *HBB* + i53, *n* = 3 for *CCR5* without i53, *n* = 5 *CCR5* + i53, *n* = 4 for *IL2RG* and *HBA1*. Mean ± SD depicted for all experiments. \**P* < 0.05 and \*\**P* < 0.005, respectively, by two-tailed paired *t*-tests (*P* = 0.002 for *HBB*; *P* = 0.0039 for *IL2RG*; *P* = 0.0271 for *HBA1*). Source data are provided as a Source Data file. **b** Indel frequencies were determined by ddPCR, ICE, or TIDE analysis 2 days post-electroporation. Data from *n* = 7 independent biological replicates for *HBB* targeting without i53, *n* = 5 *HBB* + i53, *n* = 3 for *CCR5* without i53, *n* = 5 *CCR5* + i53, *n* = 4 for *IL2RG*. Mean ± SD depicted. \*\**P* < 0.005 (*P* = 0.0063 for *HBB*) by two-tailed paired *t*-test. Source data are

provided as a Source Data file. **c** Representative FACS plots of biallelic targeting using HBB-mCherry and HBB-GFP encoding AAV donors. Representative FACS plot from *n* = 3 biologically independent experiments. **d** Various cell types (Airway basal cells, T cells, and MSCs) were gene-edited with or without i53 peptide, HDR frequencies in airway stem cells were determined by ICE or TIDER analyses. HDR in MSCs were determined by the read out of GFP expressing cells via flow cytometry. In airway basal cells, *CFTR* locus was targeted to correct and restore ΔF508[11]; in T cells, *TRAC* locus was targeted to integrate a CD19-specific chimeric antigen receptor inframe[37]; in MSCs, *HBB* locus was targeted to integrate a GFP reporter transgene[38]. All analyses were conducted 2–3 days post-electroporation unless noted otherwise. Data from *n* = 6 independent biological replicates for airway stem cells, *n* = 3 for T cells, and *n* = 3 for MSCs. Mean ± SD depicted.

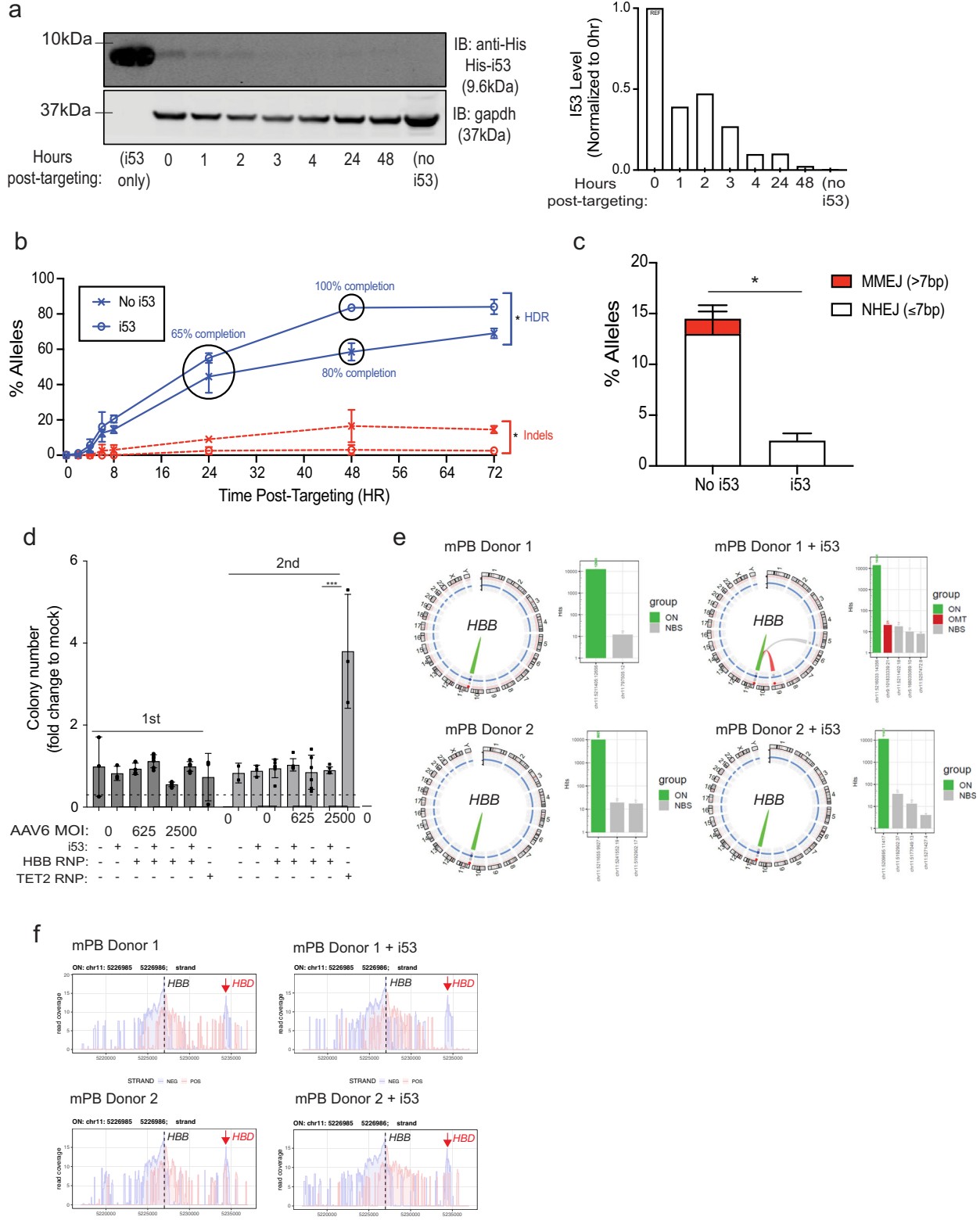

or without i53 peptide, and cells were collected at various timepoints post-editing (0, 2, 4, 6, 8, 24, 48, and 72 h). We found that, i53 peptide-treated cells completed HDR by 48 h post-editing whereas with no treatment the cells were at 80% completion with a lower average HDR frequency (Fig. 4b). Furthermore, our downstream ddPCR analysis suggests that indel formation is decreased by i53 peptide for approximately 24 h (Fig. 4c), allowing DNA repair pathway to be biased towards HDR only in the presence of AAV donor, while not permanently inhibiting the long-term function of DNA repair processes (Supplementary Fig. 3a, b). In the absence of AAV donor, we found that i53 affected the distribution of indels at the Cas9 cut site. Without the addition of i53 peptide, 25.5% of indels were insertions or deletions ≤7 bp in lengths and 46.5% of the indels were >7 bp. However, with addition of i53 peptide, only 7.5% of indels were ≤7 bp while 62% were

**Fig. 4 | Validating kinetics and safety of i53 peptide during genome editing in CD34⁺ HSPCs. a** Representative immunoblot showing the expression of His-i53 peptide in CD34⁺ cells at 0, 1, 2, 3, 4, 24 and 48 h post-electroporation. GAPDH served as a loading control. Quantification of i53 expression depicted. Three independent biological experiments conducted, and one representative blot shown. **b** HSPCs were electroporated with Cas9-RNP targeting *HBB* gene and AAV6 with or without i53 peptide. Cells were collected at timepoints 0, 2, 4, 6, 8, 24, 48, and 72 h post-electroporation and HR and indel rates were determined by ddPCR or ICE. Data conducted using HSPCs from two CB donors (*n* = 2 biological donors). For each donor, the experiment was independently conducted two times. HR represented in blue and indels are represented in red. Mean ± SD depicted. *P < 0.05 (*P* = 0.0359 for HDR; *P* = 0.0453 for indels) by two-tailed paired *t*-test. **c** Indels consisting of NHEJ and MMEJ are represented. Data from *n* = 2 biological replicates with two technical replicates. Mean ± SD depicted. *P < 0.05 (*P* = 0.0307) by two-tailed unpaired *t*-test. **d** Edited HSPCs were plated on methylcellulose the number of colonies were scored 14 days after plating and cells were replated for a second round. The results show fold changes in the number of each colony normalized to control (mock treated HSPCs) from the first round of plating. Data from *n* = 3 independent biological replicates for mock and mock + i53 treated cells and *TET2* KO cells. Data from *n* = 7 for *HBB* targeted cells (625 and 2500 AAV MOI). Mean ± SD depicted. ***P < 0.05 (*P* = 0.0009) by two-tailed unpaired *t*-test (only conducted for *TET2* RNP and *HBB* RNP+ i53 + 2500 MOI AAV). **e** Circos plots and bar graphs illustrate CAST-Seq results highlighting *HBB* target site. Lines on Circos plot represent chromosomal rearrangements with the *HBB* target site: On-target aberrations in green, OMTs with in red, and NBS in gray. Bar graphs illustrate the number of hits for different clusters of aberrations. Data from *n* = 2 biological donors with two technical replicates illustrated. **f** Coverage plots show chromosomal position vs number of reads on Chr11 around the on-target site for all samples. The large peak on the dashed line represents on-target aberrations on *HBB* and the second peak on the right represent large chromosomal deletions between *HBB* and *HBD*. **a**–**e** Source data are provided as a Source Data file.

>7 bp (Supplementary Fig. 3b). This shift might be due to a preferential effect on the NHEJ pathway choice, which typically yields indels ≤7 bp, and a bias toward alternative mechanisms of DNA damage repair such as microhomology-mediated end joining (MMEJ), which is known to yield a broader indel spectrum[39,40].

Inhibiting 53BP1 for prolonged periods would put cells at risk of genotoxicity and in fact, mice with knockouts of 53BP1 are susceptible to lymphomas derived from the hematopoietic system[28,29]. Therefore, we assessed whether an extremely short presence of the i53 peptide during genome editing associates with any aberrant expansion of blood cell clones derived from HSCs. We performed CFU assay consisting of two rounds of serial plating in methylcellulose, using HSPCs edited and transduced with AAV6 with MOI at either 625 or 2500 that either have or have not been treated with i53 peptide (Fig. 4d). Following electroporation with RNP that targets the *HBB* locus with or without i53 peptide, HSPCs were transduced with AAV6 and plated in methylcellulose media. As a positive control for increased serial plating, we used RNP that disrupts the endogenous *TET2* locus in HSPCs as it has been previously demonstrated that these cells exhibit key features of preleukemia/clonal hematopoiesis[41]. Fourteen days after plating colonies were assessed and we found that the inclusion of i53 had no discernible effect on colony-forming ability of HSPC (Fig. 4d). Additionally, upon replating, we observed no significant number of colonies with the inclusion of i53 peptide. There was a distinct advantage for TET2 knockout cells upon replating as previously reported (Fig. 4d)[41].

Another possible concern with transient inhibition of 53BP1 is that it could lead to large-scale aberrant genomic events, such as translocations. To address this concern, we employed two independent assays to probe for aberrant genomic events. We first measured translocation frequency by using a combination of two sgRNAs that target HSPCs simultaneously at *HBB* and *AAVS1* and quantified the enrichment of translocations using droplet digital PCR (ddPCR) assay as previously described[7] (Supplementary Fig. 3c). Following editing, we found mean translocation frequencies of 0.45% and 0.48% for cells treated with or without i53, suggesting no evidence for an increased frequency of translocations caused by transient expression of i53 peptide (Supplementary Fig. 3c).

Additionally, we used the highly sensitive CAST-Seq assay to characterize chromosomal aberrations at the on-target site as well as chromosomal translocation events with off-target sites, occurring in cells edited with the addition of i53. CD34+ HSPCs of two different donors were cultured in cytokine-supplemented media for 2 days, at which point the 500 K cells were collected and underwent gene editing by electroporating with RNP targeting *HBB* with or without i53 peptide treatment (no AAV6 donor was used to maximize the potential off aberrant events being detected). At day 4, electroporated cells were harvested and subjected to CAST-Seq to evaluate the dynamics of

chromosomal alterations. CAST-Seq analyses were performed as previously described using two decoy primers and a bait primer on the telomeric side[42]. Chromosomal rearrangements were classified as off-target-mediated translocation (OMT) or homology-mediated translocation (HMT). When neither an off-target nor a homology stretch could be identified, the chromosomal aberration was considered to be prompted by a non-classified break site (NBS). On-target activity, as measured by indel formation, was in the range of 60-70% (Supplementary Fig. 3e). The analysis showed that the CD34+ HSPCs treated with or without i53 peptide did not harbor any major difference in chromosomal rearrangement events (Fig. 4e, f and Supplementary Fig. 3f). Specifically, the profile of on-target mediated aberrations at the target site on Chr11 are similar in all samples treated with or without i53 peptide (Fig. 4e, f and Supplementary Fig. 3e). We confirmed only a single OMT in CD34+ HSPCs derived from donor 1 and this site is located in Chr9: 101,833,338–101,833,850, which has been previously identified using several methodologies for off-target identification[5,32,43] (Fig. 4e and Supplementary Fig. 3e). In the second donor, only a few NBS (with low number of hits) were identified on chromosome 11, likely representing large deletions/inversions with the on-target site (Supplementary Fig. 3f). Additionally, in all samples, the previously reported *HBD*[32] (which shares high sequence homology to *HBB* and is also close to the gene) off-target site was detected, as shown in the coverage plots (Fig. 4f). Overall, we adapted CAST-Seq to identify Cas9-induced chromosomal rearrangements and successfully classified them in OMT- and NBS-induced aberrations. Importantly, we did not identify any new or increased chromosomal aberrations when incorporating i53 peptide to edit CD34+ HSPCs.

## Transient inhibition of 53BP1 results in improved engraftment of gene edited cells

Because colony-forming potential is only one measure of HSPC function and the CFU assay does not analyze the presence of HSCs with self-renewal and multilineage capacity, we transplanted HSPCs into sub-lethally irradiated adult immunodeficient non-obese diabetic (NOD)-severe combined immunodeficiency (SCID)-Il2Rg⁻/⁻ (NSG) mice. Transplantation of HSPCs, either wild-type or gene-edited, into NSG mice and assessing their ability to engraft and repopulate their bone marrow would allow for the evaluation of the efficacy and safety of the gene-edited HSCs. NSG mice are highly supportive of human engraftment and hematopoietic repopulation and thereby we evaluated the in vivo engraftment ability of HSPCs targeted with low, medium, and high MOIs (625 vs. 2500 vs. 5000) of AAV6 with or without i53 peptide treatment. We edited mobilized peripheral blood (mPB) CD34⁺ HSPCs derived from four healthy donors (donors A, B, C and D) in four independent transplantation experiments (Fig. 5a). 48 h after editing, gene-targeted HSPCs were transplanted by retro-orbital injection into NSG recipient mice

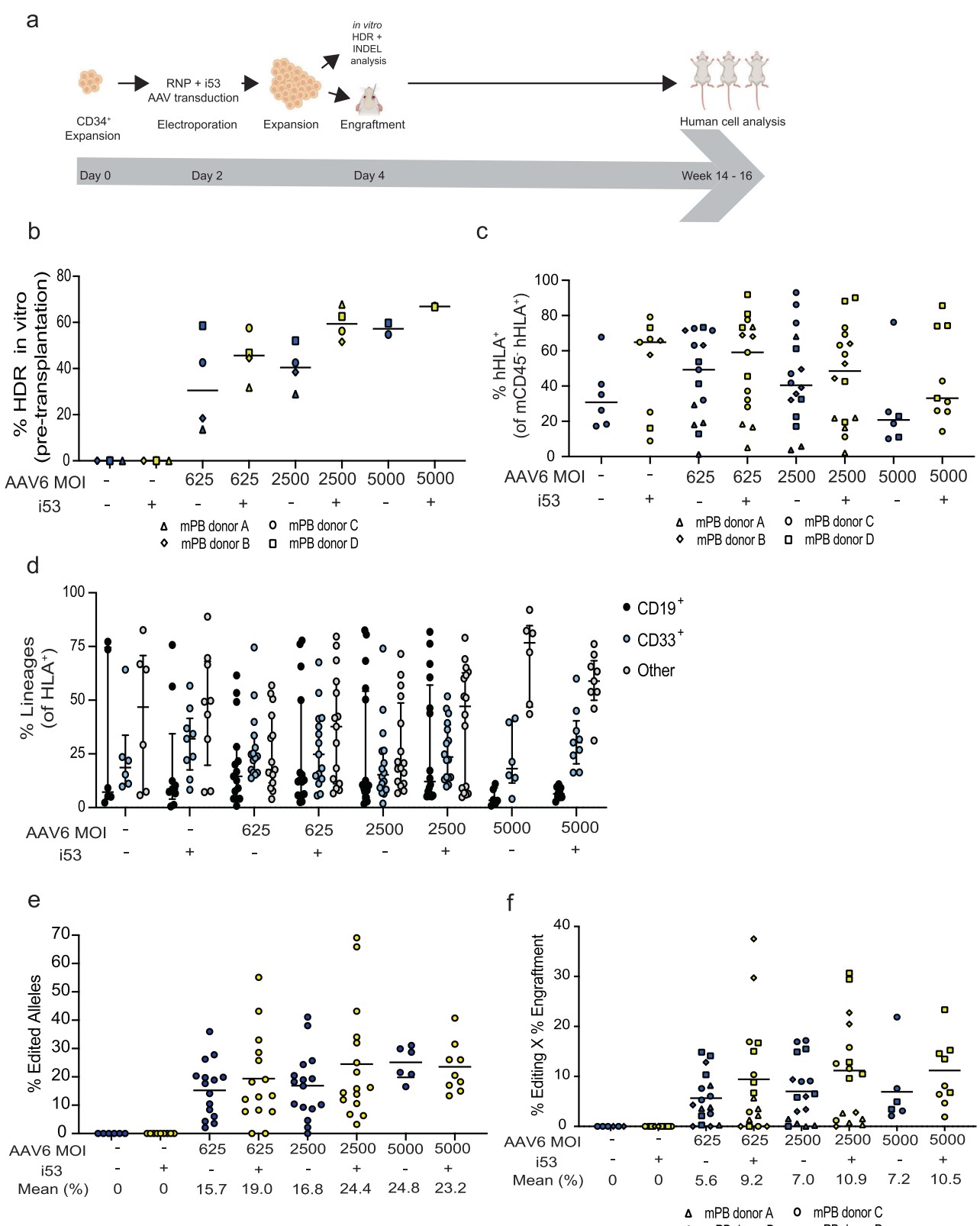

(1 × 10$^5$ cells per mouse). Subsequently, we tracked human donor cell engraftment in the recipients at week 14 or 16 (Fig. 5a). We hypothesized that transplantation of 1 × 10$^5$ HSPCs per mouse would provide a low dose of HSCs to allow us to observe meaningful differences in the functional HSCs that had been gene targeted at the *HBB* locus. Cells obtained from four donors (donors A, B, C, and D) were edited ex vivo and after confirming that the incorporation of i53

peptide improved editing frequencies at all three MOIs tested (MOIs = 625, 2500 and 5000) (Fig. 5b). Gene-edited HSPCs were transplanted as described (Fig. 5a, b). Sixteen weeks post-transplantation, we assessed the engraftment of the edited cells by measuring HLA-A/B/C as a marker in the bone marrow of all transplanted mice and found that all treatment groups were able to successfully engraft into the bone marrow (Fig. 5c, d and Supplementary Fig. 4a).

**Fig. 5 | *HBB*-gene targeted CD34⁺ HSPCs display improved long-term multi-lineage reconstitution in NSG mice. a** Experimental layout. Figure created with Biorender.com **b** *HBB* gene editing outcomes in CD34⁺ HSPCs in vitro. Data from four biological donors (donors A, B, C and D). Median values depicted. **c** Human engraftment by measuring HLA-A/B/C as a marker at 16 weeks post-transplantation in NSG mice from all experimental groups. Data from four bone marrow donors (donors A–D) represented in a combined panel. Median values reported. *n* = 6 for RNP only without i53; *n* = 9 for RNP only with i53; and *n* = 15 for RNP + 625 MOI AAV without i53; *n* = 15 for RNP + 625 MOI AAV with i53; *n* = 16 for RNP + 2500 MOI AAV without i53; *n* = 14 for RNP + 2500 MOI with i53; *n* = 6 for RNP + 5000 MOI AAV without i53; and *n* = 9 for RNP + 5000 MOI AAV with i53. **d** Among engrafted human cells, distribution among CD19+ (B cell), CD33+ (myeloid) and other (that is, HSPC/RBC/T/NK/Pre-B lineages). Bars represent median. Values represent biologically independent transplantations. **e** Total HDR alleles in human cells in the bone marrow of NSG mice. Mean values reported. **f** Percentage of genome-edited cells that have successfully engrafted in the bone marrow of NSG mice. Calculated by % editing x % engraftment. Mean values reported. **d–f** Note: *n* = 6 for RNP only without i53; *n* = 9 for RNP only with i53; and *n* = 15 for RNP + 625 MOI AAV without i53; *n* = 15 for RNP + 625 MOI AAV with i53; *n* = 16 for RNP + 2500 MOI AAV without i53; *n* = 14 for RNP + 2500 MOI with i53; *n* = 6 for RNP + 5000 MOI AAV without i53; and *n* = 9 for RNP + 5000 MOI AAV with i53. **b–f** Source data are provided as a Source Data file.

We found that engraftment was negatively impacted by the AAV6 treatments, yielding lower engraftment especially when using higher MOIs of AAV6 as previously reported (Fig. 5c and Supplementary Fig. 4b)[5,35]. Therefore, we reduced the MOI of AAV6 loaded onto HSPCs to as low as 625 MOI from 5000 MOI in order to mitigate this negative effect. We confirmed that the addition of i53 peptide does not inhibit engraftment of HSPCs and in fact, it mildly improves the engraftment in each treatment across four independent experiments compared to the standard protocol (Fig. 5c and Supplementary Fig. 4b). Overall, we confirmed that treating the cells with i53 peptide while reducing the dose of AAV6 does not jeopardize the engraftment and the lineage reconstitution potentials of edited cells (Fig. 5c, d and Supplementary Fig. 4b). We showed that the gene editing process using i53 peptide does not influence the multilineage potential of the gene-edited HSCs (Fig. 5d). Upon analyses of B cell and myeloid lineages, the contribution to different lineages was similar if not better for 625 MOI and 625 MOI + i53 peptide groups. We found no difference in the percent of gene targeted alleles following transplantation using i53 (Fig. 5e). When we combine the overall level of human cell engraftment (Fig. 5c) with frequency of gene targeted alleles (Fig. 5e), we find that the overall engraftment of total gene targeted alleles increases (Fig. 5f).

Complementing the engraftment studies we demonstrated multilineage engraftment of HDR genome edited cells, we also demonstrated that the RNP/AAV6 genome editing system could edit at high frequencies by homologous recombination all of the multiple CD34+ phenotypic sub-types that have been described including long-term HSCs (LT-HSCs)[44] (Supplementary Fig. 4f, g).

## Discussion

The ability to introduce custom edits into patient-derived HSPCs by Cas9/AAV6-mediated genome editing holds great promise for the treatment of monogenic blood disorders and other diseases. However, many challenges remain as long-term repopulating HSCs (LT-HSCs) have been found to be more resistant to HR, which traditionally has been accounted for by increasing AAV6 MOI. Additionally, it has been thought that improvement of HR frequencies in vitro would directly correlate with improvement of HDR frequencies in vivo. However, while our data indicates that increasing AAV6 load does lead to improvement of HDR frequencies in vitro, this comes at the expense of the reconstitution potential of edited HSPCs in vivo by diminishing the LT-HSC functions due to high AAV6 MOIs impairing cell fitness[5,15,45]. We show that AAV6 transduction can indeed trigger a strong p53 transcriptional response in HSCs in a AAV6 dose-dependent manner, characterized by p21 upregulation as previously shown[15,28]. We further demonstrate that the strong p53 transcriptional response can be mitigated by delivery of i53 peptide resulting in a non-toxic, kinetically rapid, and effective means of increasing HDR in human HSPCs. Our data demonstrate the use of i53 peptide during gene editing improves HDR in HSPCs and in additional human primary cells including MSCs and airway stem (basal) cells. We found that using i53 peptide gave high frequencies of HR with lower AAV6 MOIs while reducing p21 activation following the editing process, To ensure safety, we show that i53 peptide is only transiently present in HSPCs and does not lead

to increased translocations, Most importantly, we highlight that it is possible to preserve HSC function and improve the total number of HDR edited alleles engrafted in vivo by incorporating i53 into the genome editing process.

Schiroli et al. have previously shown that the induction of DSBs following gene editing activates a p53 transcriptional response in HSPCs that is further aggravated by transduction of AAV6 donor templates[15]. Consequently, these effects may diminish the ability of HSCs to renew and generate multilineage cells and further reduce their engraftment potential[46]. To overcome these challenges, Lehnertz et al. showed that transient *TP53* knockdown significantly enhanced the recovery of HSCs[47] and Ferrari et al. developed an enhanced gene editing protocol using GSE56, an inhibitor of p53 activation to ameliorate the negative impact on edited LT-HSCs[46]. Here, we propose an alternative strategy by incorporating a i53 peptide which inhibits 53BP1. Brault et al. demonstrated that using mRNA to express i53 and an mRNA to inhibit p53 directly that they could achieve better genetic engineering frequencies and better engraftment compared to lentiviral modification at the IL2RG locus[48]. In this work, we have identified that i53 peptide can also be used and because of its shorter expression might have a better safety profile than i53 mRNA expression. This work is new in showing that it can be used as a peptide to increase HR and as a mechanism to lower the MOI of AAV6. Future work will test whether combining i53 peptide with p53 inhibition provides a further increase in engraftment of HDR edited cells. The use of i53 peptide allows us to significantly reduce the amount of AAV6 used, and we show that it is indeed possible to use lower MOI of AAV6 and simultaneously improve HDR and HSPC function by inhibiting the AAV6-mediated p53 transcriptional response. Therefore, the use of i53 peptide allows us to dramatically improve in the ability of HSCs to form CFU colonies.

Because the majority of i53 peptide is rapidly degraded within 4 h post-electroporation, 53BP1 suppression is likely to be transient and would not lead to long-term disruption of the normal DNA damage response. While we did not find i53 delivery to increase frequency of large-scale genomic aberrations, further investigation to thoroughly assess the genotoxicity and the long-term safety of transient 53BP1 inhibition will be needed prior to clinical translation[42].

In conclusion, not only does the delivery of i53 peptide improve editing frequencies at a variety of loci in a variety of primary cell types, but also allows high levels of editing to be attained with lower amounts of AAV6. With a significantly lower MOI of AAV6, it is biologically plausible that the already low frequency of random integrations of the vector genome would decrease further and thereby improve the safety of this approach[49,50]. Consequently, we found that i53 improves cell viability, differentiation potential of HSPCs, and the ability of HSCs to engraft long-term in NSG mice. Moving forward, these data suggest that i53 might be safely integrated into the standard CRISPR/AAV6-mediated genome editing protocol to attain greater numbers of edited cells for transplantation of clinically meaningful cell populations.

# Methods

## CD34+ HSPCs

Frozen CD34+ HSPCs derived from mobilized peripheral blood (mPB) were purchased from AllCells and thawed according to manufacturer's instructions. CD34+ HSPCs from cord blood were either purchased frozen from AllCells or acquired from donors under informed consent via the Binns Program for Cord Blood Research at Stanford University and used fresh without freezing[51]. The works was performed on blinded samples from healthy donors of CD34+ HSPCs and inclusive of all cells that were donated to research. The research was performed, analyzed and communicated in an ethical manner.

Cord blood-derived CD34+ cells were used for all in vitro experiments. unless noted otherwise. mPB derived CD34+ HSPCs were used for all transplantation experiments shown in Fig. 5 and in vitro experiments shown in Fig. 2c−e. All CD34+ HSPCs were cultured in GMP SCGM medium (Cellgenix) supplemented with SCF (100 ngml$^{-1}$), TPO (100 ngml$^{-1}$), Flt3 ligand (100 ngml$^{-1}$), IL-6 (100 ngml$^{-1}$), and StemRegenin1 (0.75 mM). Cells were cultured at 37 °C, 5% CO$_2$, and 5% O$_2$.

## Mesenchymal stromal cells (MSCs)

MSCs were cultured as previously described[38]. Briefly, frozen hBM-MSCs from healthy donors were obtained from AllCells Inc and were thawed according to standard protocols. Cells were cultured at a density of 60 cells/cm$^2$ in complete culture medium consisted of α-modified minimum essential medium (α-MEM, Gibco, Invitrogen, Carlsbad, CA), supplemented with 16.7% (v/v) fetal bovine serum (Gibco), 2mM L-glutamine, and 1% penicillin/streptomycin cocktail. Cells are subcultured when confluency reached 80−90%.

## T cells

T cells were cultured as previously described[52]. Briefly, T cells were isolated from buffy coats that were obtained from healthy donors at the Stanford Blood Center with isolation of peripheral blood mononuclear cells (PBMCs) on a Ficoll density gradient followed by magnetic enrichment using a Pan T Cell Isolation kit (Miltenyi). T cells were cultured in X-Vivo 15 with gentamicin (Lonza), 5% human serum (Sigma-Aldrich) and 100 IU ml$^{-1}$ human IL-2 (Peprotech). Cells were cultured at 37 °C with 5% CO$_2$ and ambient O$_2$. The culture medium was refreshed every 2−3 days.

## Airway basal stem cells

Airway basal stem cells were isolated and cultured as previously described[11]. Briefly, KRT5+ (KRT5, Abcam, ab 193895) cells were isolated from upper airway tissue was obtained from adult patients. KRT5+ cells were plated at a density of 10,000 cells per cm$^2$ on tissue culture plates coated with 5% BME. Cells were incubated at 37 °C in 5% O$_2$ and 5% CO$_2$ in EN media with 10 μM ROCK inhibitor (Y-27632, Santa Cruz, sc-281642A). EN media consists of ADMEM/F12 supplemented with B27 supplement, nicotinamide (10 mM), human EGF (50 ng/mL), human Noggin (100 ng/mL), A83-01 (500 nM), N-acetylcysteine (1 mM) and HEPES (1 mM).

## AAV vector production

rAAV6 was produced as previously described[5]. In brief, HEK-293T (purchased from ATCC CRL3216) cells were cotransfected using polyethylenimine (PEI) with the pDGM6 helper plasmid and the respective transfer plasmid. After harvesting cells, cell lysates were treated with Benzonase (Sigma-Aldrich), the crude AAV extract was purified on an iodixanol density gradient and dialysis was performed against PBS/sorbitol. AAV titer was determined by measuring the absolute concentration of inverted terminal repeat copy numbers by droplet digital PCR (Bio-Rad) according to the manufacturer's protocol using previously reported primer and probe sets[53].

## Genome editing of CD34+ HSPCs, MSCs, T cells and basal cells

Chemically modified sgRNAs used to edit CD34+ HSPCs were purchased from Synthego and were purified by HPLC. The sgRNA modifications added were 2'-O-methyl-3'-phosphorothioate at the three terminal nucleotides of the 5' and 3' ends. All Cas9 protein (Alt-R S.p. Cas9 Nuclease V3) used was purchased from Integrated DNA Technologies. RNPs were complexed at a Cas9/sgRNA molar ratio of 1:2.5 at 25 °C for 10 min before electroporation. CD34+ cells were resuspended in P3 buffer (Lonza) with complexed RNPs. I53 peptide is added directly to the RNP and cell mixture immediately before electroporation. Electroporation was carried out using the Lonza 4D Nucleofector (program DZ-100 for HSPCs, and T cells; CM-119 for MSCs; CA-137 for airway basal cells). Cells were plated at $2.5 \times 10^5$ cells/ml following electroporation in the cytokine-supplemented media described previously. Immediately following electroporation, AAV6 was supplied to the cells at indicated (MOI range 625−5000) vector genomes per cell, based on titers determined using a Bio-Rad QX200 ddPCR machine and QuantaSoft software (v.1.7; Bio-Rad). For genome editing of T cells, 3 days before electroporation, T cells were activated with anti-CD3/anti-CD28 paramagnetic beads (Dynabeads Human T Cell Activator, Gibco). Activation beads were removed before electroporation.

## Gene-targeting analysis by flow cytometry

Between 4 and 8 days post targeting with fluorescent gene replacement vectors, targeted cells were collected and the percentage of edited cells was determined by flow cytometry. Reporter expression (GFP or mCherry expression) was assessed using FACS Aria II cytometer and FACS Diva software (v.8.0.3; BD Biosciences). The data were subsequently analyzed using FlowJo (v.10.6.1; FlowJo LLC).

## Indel frequency analysis by ICE/TIDE

Between 2 and 4 days post targeting, gene edited cells were harvested and QuickExtract DNA extraction solution (Epicentre) was used to collect gDNA. INDEL frequencies were quantified using the TIDE/ICE software (tracking of indels by decomposition) and sequenced PCR products obtained by PCR of genomic DNA extracted after electroporation as previously described[36].

## Genomic editing frequencies by droplet digital PCR (ddPCR)

Droplet digital PCR (ddPCR) was used to quantify genomic editing frequencies using genomic DNA extracted using QuickExtract DNA Extraction Solution (Epicenter). Quantification of HBB gene editing was performed using the primer/probe by ddPCR. HBB ddPCR forward: **tcactagcaacctcaaacagac**; HBB ddPCR reverse: **cctgtcttgtaaccttgatacc**. Briefly, genomic DNA was extracted using QuickExtract DNA Extraction Solution and PCR amplicons (length of 1410 bp) spanning the targeted region were generated using the HBB in-out primer pair. HBB out forward: **aggaagcagaactctgcacttca**; HBB in reverse: **agtcagtgcctatcagaaacccaagag**. PCR cycling conditions for in-out PCR are as follows: 98 °C (30 s); followed by 35 cycles of 98 °C (10 s), 60 °C (30 s), and 72 °C (1 min); and finally 72 °C (10 min). The PCR products were run on a 1% agarose gel and the band located at 1410 bp was cut and purified using QIAquick gel extraction kit (Qiagen). Subsequent PCR product was diluted to 10 ng/μl. Subsequently, six serial dilutions of the PCR products were made in nuclease-free H2O down to 10−20 fg/μl, which served as the template DNA for the ddPCR reaction. ddPCR reactions were set up using a HBB ddPCR primer pair and HR-(HEX), REF-(HEX), and WT-(FAM) probes. HR probe (HEX): **tgactcctgaggaAaaAtcCgcAgtCa**; Reference probe (HEX): **acgtggatgaagttggtggtgagg**; and WT probe (FAM): **ccccacagggcagtaacggcagacttc**. Droplets were generated and analyzed according to manufacturer's instructions using the QX200 system (Bio-Rad). ddPCR cycling conditions were as follows: 98 °C (10 min);

followed by 50 cycles of 94 °C (30 s), 60 °C (30 s), and 72 °C (2 min); and then 98 °C (10 min).

## P21 quantification by ddPCR
Droplet digital PCR (ddPCR) was used to quantify the expression of p21 from RNA extracted using RNeasy plus micro kit (Qiagen; 74034). Extracted RNA was reverse transcribed using iScript Reverse Transcription Supermix (Bio-Rad; 1708841) and 6 uL of reversed transcribed cDNA was used to assess the levels of p21 and ACTB in a total reaction volume of 20 uL. P21 expression was normalized to ACTB. ddPCR probes: CDKN1A (FAM) (Bio-Rad; dHsaCP2500375) and ACTB (HEX) (dHsaCPE5190200). ddPCR cycling conditions were as follows: 98 °C (10 min); followed by 40 cycles of 94 °C (30 s), 60 °C (1 min); and then 98 °C (10 min).

## Immunoblot
Cells were harvested and washed with PBS. Whole-cell extract was prepared from 1X radioimmunoprecipitation assay lysis buffer (RIPA; Millipore). Extract was clarified by centrifugation at 15,000 $g$ for 15 min at 4 °C, and protein concentration was determined by Pierce BCA (bicinchoninic acid) assay (Thermo Fisher Scientific). 8–30 μg of whole-cell extract was separated on precast 4 to 12% bis tris protein gel (Invitrogen) and transferred to a nitrocellulose membrane. Membrane was blocked in PBS–0.05% Tween 20 (PBST) containing 5% nonfat dry milk and incubated overnight at 4 °C with primary antibody diluted in PBST–5% nonfat dry milk or 5% bovine serum albumin (BSA, Sigma). Membranes were subsequently washed with PBST and incubated with the appropriate IRDye 680RD and IRDye 800CW secondary antibody (LI-COR Biosciences) diluted in PBST–5% nonfat dry milk. Images were detected using the Odyssey Systems (LI-COR Biosciences). The following primary antibodies were used: His-tag (2365), GAPDH (2118S) purchased from Cell Signaling (Cell Signaling Technology). Full scan blots are provided in the Source Data file.

## Analysis of HBB-AAVS1 translocations
HBB-AAVS1 translocations were measured as previously described[7]. Briefly, for ddPCR quantification of translocations, a HEX reference assay detecting copy number input of the *TERT* gene was used to normalize for genomic DNA input (Bio-Rad: saCP1000100). A custom assay designed to detect the translocations between *HBB* and *AAVS1* consisted of: Forward primer: 5′-TCAGGGCAGAGCCATCTATTGC-3′, Reverse primer: 5′-CCAGATAAGGAATCTGCCTAACAGG-3′, 5′-6FAM/ZEN/3′-IBFQ-labeled Probe (IDT): 5′-CTTCTGACACAACTGTGTTCACTAGCAACC-3′.

## CAST-Seq
CAST-Seq analyses were performed as previously described[42] but using an improved algorithm to classify chromosomal rearrangements[54]. Two decoy primers and bait primers on the telomeric side were used: Decoy 1, 5′-CAGGTTGGTATCAAGGTTACAAG; Decoy 2, 5′-AACTTCATCCACGTTCACC; Bait, 5′-TGCTTCTGACACAACTGTGT, and Bait Nested, 5′-CACAACTGTGTTCACTAGCAACCTC. The generated libraries were sequenced on a NovaSeq using 2 × 150 bp paired-end sequencing (GENEWIZ). For every sample, two technical replicates were performed. Shown are events that were significant in both runs, if not indicated otherwise.

## Immunophenotyping of HSPCs
Cells were stained with LIVE/DEAD Fixable Blue Dead Cell Stain (Life Technologies) and then with anti-human CD34 PE-Cy7 (581, BioLegend; 1:100), CD38 Alexa Fluor 647 (AT1, Santa Cruz Biotechnologies; 1;50), CD45RA BV 421 (HI100, BD Biosciences; 1:25), and CD90 BV605 (5E10, BioLegend; 1:30) and analyzed by flow cytometry. For sorting of CD34+ or CD34+ CD38− CD90+ cells, cord-blood-derived CD34+ HSPCs were stained directly after isolation from blood with anti-human CD34

FITC (8G12, BD Biosciences; 1:100), CD90 PE (5E10, BD Biosciences; 1:50), CD38 APC (HIT2, BD Bioscience; 1:50), and cells were sorted on a FACS Aria II (BD Bioscience).

## Methylcellulose CFU assay
The CFU assay was performed by plating 300–900 cells on plate containing MethoCult Optimum (StemCell Technologies) 4 days after electroporation and/or transduction. After 12–16 days, colonies were counted and scored based on their morphological appearance in a blinded fashion using STEMvision (StemCell Technologies). For serial plating, after counting the colonies, cells were harvested and replated with 100,000 cells/plate.

## CD34+ HSPC transplantation into immunodeficient NSG mice
Six- to eight-week-old female NSG mice (Jackson Laboratory) were irradiated using 200 rad, 12–24 h before transplantation, with targeted HSPCs (2 d post targeting) via retro-orbital injection. Approximately $1 \times 10^5$ electroporated HSPCs were injected per mouse using an insulin syringe with a 27-G, 0.5-inch (12.7-mm) needle. Mice were housed at an ambient temperature of 22 °C with 50% humidity and a 12/12-h light/dark cycle. This experimental protocol was approved by Stanford University's Administrative Panel on Laboratory Animal Care. All mouse studies reported in this paper were performed as a minimum of three separate experimental replicates of editing and transplantation. For sample size, we transplanted as many mice as feasible to cover the non-Gaussian distribution that would be expected from experimental and donor variability, while also minimizing the total number of animals as per the FDA Center for Biologics Evaluation and Research guidelines.

## Assessment of human engraftment
Between 14 and 16 weeks post transplantation of edited CD34+ HSPCs, mice were euthanized and bone marrow was harvested from tibia, femur, pelvis, sternum and spine using a pestle and mortar. Mononuclear cells (MNCs) were enriched using a Ficoll gradient centrifugation (Ficoll-Paque Plus, GE Healthcare) for 25 min at 2000 $g$ at room temperature. The samples were then stained for 30 min at 4 °C with the following antibodies: monoclonal anti-human CD33 BV 421(1:50 dilution, 6 ul in 300 ul of MNCs pelleted in MACS buffer (1× PBS, 2% fetal bovine serum, 2 mM EDTA); anti-human HLA-ABC FITC (1:100 dilution; W6/32; BioLegend); anti-human CD19 PerCp-Cy5.5 (1:20 dilution; HIB19; BD Biosciences); anti-mouse CD45.1 PE-Cy7 (1:200 dilution; A20; eBiosciences); anti-human CD34 APC (1:50 dilution; 581; BioLegend); Multilineage engraftment was established by the presence of myeloid cells (CD33+) and B cells (CD19+) in engrafted human cells (mCD45-, HLA-A/B/C+ cells).

## Reporting summary
Further information on research design is available in the Nature Portfolio Reporting Summary linked to this article.

# Data availability
CAST-seq data generated in this study have been deposited in NCBI's Gene Expression Omnibus and are accessible through GEO Series accession number GSE246766. All reagents and protocols will be made available to researchers upon request from the corresponding author. Source Data are provided with this paper.

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

## Acknowledgements

We thank Geoffroy Andrieux for bioinformatics support with CAST-Seq analysis. This work was funded through NIH grant support R01HL135607, and the Laurie Kraus Lacob Faculty Scholar Fund in Pediatric Translational Medicine. We thank the Doris Duke Charitable Foundation through grant 2019112 (to M.H.P.) and the German Research Foundation grant CRC1160-A07 (to T.C.) for funding support. We thank the Binns Family Program for Cord Blood Research for providing umbilical cord blood-derived CD34+ cells. M.H.P. is supported by the Sutardja Chuk Professorship in Definitive and Curative Medicine.

## Author contributions

D.P.D. and M.H.P. supervised the project. R.B., M.K.C., D.P.D., and M.H.P. designed experiments. R.B. and M.K.C. carried out experiments. S.E.G. (IDT) and C.V. (IDT) performed the purification of i53 recombinant peptide. S.S. performed the irradiation of NSG mice. K.O.C., J.K., and T.C. performed and interpreted CAST-Seq analysis. A.M.D. and W.N.F. performed experiments related to engraftment studies. R.B., M.K.C., D.P.D. and M.H.P. wrote the manuscript. S.E.G. and C.V. contributed in analyzing and interpreting data as well as reviewing the manuscript. IDT should be contacted if an investigator wishes to test the i53 peptide.

## Competing interests

The authors of this study also wish to declare the following conflicts of interest: M.H.P. is a member of the scientific advisory board of Allogene Therapeutics. M.H.P. is on the Board of Directors of Graphite Bio. M.H.P. has equity in CRISPR Tx. M.K.C. and M.H.P. have equity in Graphite Bio. D.P.D. was an employee of Graphite Bio. C.A.V. and S.E.G. are employees of Integrated DNA Technologies, Inc. The remaining authors declare no competing interests.

## Ethical approval

The works was performed on blinded samples from healthy donors of CD34+ HSPCs and inclusive of all cells that were donated to research. The research was performed, analyzed and communicated in an ethical manner.
