## [Peer Review File · Nature Communications]

Transient Inhibition of 53BP1 Increases the Frequency of Targeted Integration in Human Hematopoietic Stem and Progenitor CellsREVIEWER COMMENTS

Reviewer #1 (Remarks to the Author):

The authors present a new approach to increasing the frequency of HDR in human HSPCs by the delivery of i53 recombinant protein. It is well-designed and well-validated, expanding the current knowledge and application of HDR frequencies can be increased by inhibiting 53BP1. I just have a few comments as shown below.

1. Line 18, it's better to define the term and mechanism of "i53" for the first time
2. Line 33, there are also large deletions (kilobase or even megabase) and insertions during NHEJ. It will be good to comment on it
3. Line 100-101, besides avoiding type-I IFN activation using naked DNA, one more advantage of using i53 protein is it will reduce the integrations of plasmid DNA into CRISPR-induced DSBs which is a concern for gene editing (PMID: 32095517; PMID: 32034391)
4. As mentioned above, the authors are encouraged to comment on CRISPR-induced DSBs, besides indels and large genomic rearrangements, there are also plasmid integrations, AAV/lentiviral integrations, and retrotransposon integrations (PMID: 31570731; PMID: 35760782; PMID: 36639728). Thus, since using the i53 protein can reduce the AAV6 MOI, which means it will greatly reduce the potential side-effect of AAV integrations in the CRISPR-edited site, improving the safety of CRISPR editing
5. Line 144-145, "We found that i53 peptide increased HDR in HSPCs in a dose-dependent manner by 10%, 46%, and 38% at 250, 1500, and 5000 µg/mL, respectively (Figure 1c)." It's confusing the numbers here are different from the ones in the figure.
6. Line 196, should be "Cas9 cutting site"

Reviewer #2 (Remarks to the Author):

In this report, Baik et al described the use of i53 recombinant protein to inhibit 53BP1 during genome editing to improve the level of HDR-mediated genome editing. Increasing rates of HDR allowed use of lower AAV MOI that reduces cellular toxicity during the genome editing process.

As noted by the authors, the use of i53 to promote HDR repair was reported previously (Canny et al., De Ravin et al., Brault et al), as is the detrimental effects of AAV (Schiroli et al,) compounded by double strand DNA breaks.

The large amount of data in vitro and transplant studies support the claim that i53 improved HDR-repair, and the use of a protein version of i53 has potential broad application for genome editing therapeutic applications.

A main challenge in interpreting the data is a lack of details for the experiments, for example, there is minimal description of the specific sgRNAs used, design of donor(s), design of donor in relation to the target sequence, and primers for evaluating HDR efficiency.

The engraftment rates are also very low, making it difficult to ascertain if there is substantive difference made by the addition of i53.

Please clarify the following points.

1. L119 As the authors indicated, 53BP1 knockout mice are at risk of lymphomas. What is the risk of lymphomas or other genotoxicities following intentional 53BP1 suppression with 53BP1 protein? Admittedly the recombinant protein is short-lasting, and the presumption is a lowering of the risks but actual data of a risk assessment would be important.
2. L125 -132: There is no description of what the cells were electroporated with. Were CRISPR/Cas9 delivered as well? Do you expect a DSB? There is no description of the AAV6 either. One presumes Cas9/sg was used to achieve HDR, but there is no mention of either, making it impossible to interpret.
3. Fig. 1c-please show details of donor design and the sequence homologies between donor and target, and primers for the ddPCR. What are the variables analyzed for statistics in Fig. 1c? ie the difference between % alleles at 500ug/ml is compared with which sample for the stats?
4. Please explain the rationale for assay measuring engraftment rates
5. Fig 5e. The percentages of edited alleles appear higher in samples corrected without i53.
6. Sequencing data to characterize the outcome of genome editing should be provided, especially in considering HDR.

Reviewer #3 (Remarks to the Author):

In this manuscript Baik et al. report that co-delivery of i53 peptide along with Cas9RNP and AAV6 donor vector increases HDR frequency at low MOI, allowing decreased toxicity of high AAV6 MOI in human HSPC. Since inhibition of 53BP1 favors repair of DSB by HR rather than NHEJ, the frequency of INDELS is decreased at targeted loci. Lowering the AAV6 MOI has a beneficial impact on CFU frequency, while a modest effect of i53 addition is observed in vivo on engraftment potential with % of in vivo edited alleles lower at low MOI. Thus, considering the author's aim at developing targeted editing in a clinically meaningful population, the biological relevance in in vivo repopulating cells is debatable. In cell types different from HSPC, such as airways stem cells, MSCs and T-cells, the addition of i53 has a variable effect on % HDR increase, with the highest in the former and the lowest in the latter.

The concept of transiently inhibiting 53BP1 to increase HDR in human HSPC is not novel, since recently reported by others. Here, the authors exploit the use of protein delivery differently from mRNA assuming that this strategy might be safer. However, no experiments of direct comparison are shown. In general, there are some inconsistencies and statements that are not fully supported by the presented data. Major comments are listed.

1. In figure 1a, 2a and 2b, the input cells for CFU assay is different. The author should be consistent among different experiments. In the legend of figure 2, is data n>4 referred to biological or technical replicates? In fig.2b % HDR at low MOI +i53 is higher (>60%) than in fig.2e, could the authors explain the data?
2. In figure 3a, the authors should annotate which are CB and mPB samples. Is there any difference about the source? The addition of i53 increase %HDR of 30-40%, but the basal level of HDR looks gene-specific. The authors should discuss the explanation for this finding. As well as different results in different cell types.
3. In figure 5 results showed no detrimental effect of i53 treatment on HSPC in vivo engraftment, with modest improvement (fig.5c) and even lower % editing at low MOI (fig5e). The title of the results section is not supported by data since no improved engraftment of edited cells is biologically meaningful. The experiments were performed by transplanting CB-CD34+ cells. The authors should repeat the experiments with a clinically relevant source, such as mPB.
4. Discussion line 293: the statement of improved HDR in vivo is not supported by these data. Line 302: ref De Ravin et al. should be cited also here and those data should be discussed. Line 307: the dramatically improved number of CFU by using i53 is not supported by data, since the graph in figure 2b lacks the control of low MOI-i53. Line 317: the claim of improved engraftment in NSG mice should be corroborated by statistically significant differences and biologically relevant

implications.

Minor:

Figure 4b: should the authors increase the number of biological replicates, at least $n=3$?

Methods: supplier for i53 should be indicated.

References: accurate check is required, some references are incomplete (#28, 41).

Response to Reviewer Comments

Dear Reviewers,

Thank you for your careful review of our manuscript entitled “Transient Inhibition of 53BP1 Increases the Frequency of Targeted Integration in Human Hematopoietic Stem and Progenitor Cells” for potential publication in *Nature Communications*. In the revised manuscript, we address several points both textually and with experimental data concerning the safety of using i53 peptide to transiently inhibit 53BP1 during genome editing. We also provide additional experimental detail as requested to help readers better understand and hopefully use our findings in their own experiments. Below we provide point by point responses to each of the comments.

Reviewer #1 (Remarks to the Author):

The authors present a new approach to increasing the frequency of HDR in human HSPCs by the delivery of i53 recombinant protein. It is well-designed and well-validated, expanding the current knowledge and application of HDR frequencies can be increased by inhibiting 53BP1. I just have a few comments as shown below.

1. Line 18, it's better to define the term and mechanism of "i53" for the first time.

Thank you and we have made the requested change.

2. Line 33, there are also large deletions (kilobase or even megabase) and insertions during NHEJ. It will be good to comment on it

We thank the reviewer for highlighting this. We have included in the text a comment that if i53 were to move into the clinic, that a more thorough assessment of genotoxicity would be needed, including whether inhibition of 53BP1 would increase such large deletions.

3. Line 100-101, besides avoiding type-I IFN activation using naked DNA, one more advantage of using i53 protein is it will reduce the integrations of plasmid DNA into CRISPR-induced DSBs which is a concern for gene editing (PMID: 32095517; PMID: 32034391)

We thank the reviewer for highlighting this additional benefit. We have included adding this benefit to the manuscript.

4. As mentioned above, the authors are encouraged to comment on CRISPR-induced DSBs, besides indels and large genomic rearrangements, there are also plasmid integrations, AAV/lentiviral integrations, and retrotransposon integrations (PMID: 31570731; PMID: 35760782; PMID: 36639728). Thus, since using the i53 protein can reduce the AAV6 MOI, which means it will greatly reduce the potential side-effect of AAV integrations in the CRISPR-edited site, improving the safety of CRISPR editing

Again, we thank the reviewer for highlighting that with a lower MOI, it is biologically plausible that the already low frequency of random integrations would decrease further and thereby improve the safety of this approach.

5. Line 144-145, "We found that i53 peptide increased HDR in HSPCs in a dose-dependent manner by 10%, 46%, and 38% at 250, 1500, and 5000 $\mu\text{g}/\text{mL}$, respectively (Figure 1c)." It's confusing the numbers here are different from the ones in the figure.

We apologize for the confusion. We have made the corrections.

6. Line 196, should be "Cas9 cutting site"

We thank the reviewer for the careful reading and we have made the correction.

Reviewer #2 (Remarks to the Author):

In this report, Baik et al described the use of i53 recombinant protein to inhibit 53BP1 during genome editing to improve the level of HDR-mediated genome editing. Increasing rates of HDR allowed use of lower AAV MOI that reduces cellular toxicity during the genome editing process. As noted by the authors, the use of i53 to promote HDR repair was reported previously (Canny et al., De Ravin et al., Brault et al), as is the detrimental effects of AAV (Schiroli et al,) compounded by double strand DNA breaks.

The large amount of data in vitro and transplant studies support the claim that i53 improved HDR-repair, and the use of a protein version of i53 has potential broad application for genome editing therapeutic applications.

A main challenge in interpreting the data is a lack of details for the experiments, for example, there is minimal description of the specific sgRNAs used, design of donor(s), design of donor in relation to the target sequence, and primers for evaluating HDR efficiency.

The engraftment rates are also very low, making it difficult to ascertain if there is substantive difference made by the addition of i53.

Please clarify the following points.

1. L119 As the authors indicated, 53BP1 knockout mice are at risk of lymphomas. What is the risk of lymphomas or other genotoxicities following intentional 53BP1 suppression with 53BP1 protein? Admittedly the recombinant protein is short-lasting, and the presumption is a lowering of the risks but actual data of a risk assessment would be important.

Thank you for sharing your safety concerns regarding the use of 53BP1 peptide. The risks associated with suppressing 53BP1 are valid concerns and therefore we believe that it is extremely important that we conduct a comprehensive investigation to assess the long-term safety of transient 53BP1 inhibition prior to clinical translation. As Reviewer #2 notes, inhibiting

53BP1 for prolonged periods would put cells at risk of genotoxicity. In fact, mice with mutation in 53BP1 will all develop cancer. Therefore, an important principle when inhibiting this pathway is to do so for as a short time as possible. And we have approached this by following a general pharmacologic principle of using the lowest and shortest dose of the drug to obtain the desired outcome. We have performed dose titrations and shown an extremely short presence of the i53 peptide to achieve the desired effect.

We note that one risk of inhibiting 53BP1 is that it might induce translocations and we demonstrate that transient inhibition does not induce higher frequencies of translocations (Figure S3d). To further explore the risks associated with transiently suppressing 53BP1 with i53 peptide, we have included a serial plating assay on methylcellulose to capture the aberrant HSC progenitor function if any (Figure 4d), and report that the inclusion of i53 has no discernible effect on colony-forming ability of HSPCs upon replating.

2. L125 -132: There is no description of what the cells were electroporated with. Were CRISPR/Cas9 delivered as well? Do you expect a DSB? There is no description of the AAV6 either. One presumes Cas9/sg was used to achieve HDR, but there is no mention of either, making it impossible to interpret.

Thank you for pointing this out and we apologize for not providing enough details to fully interpret the findings. We got caught too much in our own work and forgot the need to explain the entire experimental design. We have clarified the purpose of this experiment and included additional details describing what the cells were electroporated with and which AAV6 was used.

3. Fig. 1c-please show details of donor design and the sequence homologies between donor and target, and primers for the ddPCR. What are the variables analyzed for statistics in Fig. 1c? ie the difference between % alleles at 500ug/ml is compared with which sample for the stats?

We have included a schematic illustrating the SCD correcting AAV6 donor template spanning the Cas9 cutting site (Figure S1b). Additionally the sets of primers used for IN/OUT PCR and ddPCR have been annotated on the schematic (Figure S1b). The sequences of the primers used for IN/OUT PCR and ddPCR are also listed in Methods and a comprehensive list of all primer sequences is included Tables S2 – S4. All statistical tests were run in comparison to CD34⁺ HSPCs electroporated with Cas9-RNP and AAV6 with no i53 addition and we have added this detail to the appropriate figure legend (Figure 1c).

4. Please explain the rationale for assay measuring engraftment rates

We have added our rationale for conducting transplantation assay using gene-edited HSCPs into NSG mice (lines 247-253).

5. Fig 5e. The percentages of edited alleles appear higher in samples corrected without i53.

The reviewer has identified a finding that we have also thought deeply about. We plot and report the median because that is the correct “average” to use in this setting. Nonetheless, it is easy to see the tremendous variability between mice in Figure 5 and the mean actually being higher in the mice receiving i53 treated cells because a few mice with treated cells have much higher engraftment of HDR edited cells (now included in the figure). In part this variability reflects the stochastic and oligoclonal nature of human cell engraftment (Sharma et al Nat Comm (2021) and Ferrari et al Nat Biotech (2020)). One can also see in SF4c, the effect of variability in the source of donor cells. While the relative frequency of HDR editing is important, it is also important that the total number of edited cells that engraft is also important (hence our inclusion of Figure 5E). We wish the experimental model and its inherent heterogeneity derived most likely from the oligoclonality was better.

6. Sequencing data to characterize the outcome of genome editing should be provided, especially in considering HDR.

We have included additional sequencing data analyzed by ICE to show HDR and Indel frequencies (Figure S2a and S2b).

Reviewer #3 (Remarks to the Author):

In this manuscript Baik et al. report that co-delivery of i53 peptide along with Cas9RNP and AAV6 donor vector increases HDR frequency at low MOI, allowing decreased toxicity of high AAV6 MOI in human HSPC. Since inhibition of 53BP1 favors repair of DSB by HR rather than NHEJ, the frequency of INDELS is decreased at targeted loci. Lowering the AAV6 MOI has a beneficial impact on CFU frequency, while a modest effect of i53 addition is observed in vivo on engraftment potential with % of in vivo edited alleles lower at low MOI. Thus, considering the author’s aim at developing targeted editing in a clinically meaningful population, the biological relevance in in vivo repopulating cells is debatable. In cell types different from HSPC, such as airways stem cells, MSCs and T-cells, the addition of i53 has a variable effect on % HDR increase, with the highest in the former and the lowest in the latter.

The concept of transiently inhibiting 53BP1 to increase HDR in human HSPC is not novel, since recently reported by others. Here, the authors exploit the use of protein delivery differently from mRNA assuming that this strategy might be safer. However, no experiments of direct comparison are shown. In general, there are some inconsistencies and statements that are not fully supported by the presented data. Major comments are listed.

1. In figure 1a, 2a and 2b, the input cells for CFU assay is different. The author should be consistent among different experiments. In the legend of figure 2, is data n>4 referred to biological or technical replicates? In fig.2b % HDR at low MOI +i53 is higher (>60%) than in fig.2e, could the authors explain the data?

We apologize for the confusion and thank you for pointing out the inconsistency in the input numbers in between experiments. To clarify, we conducted all CFU assays with 300 input cells. We initially thought it would be ideal to combine our technical replicates when reporting our

numbers (and therefore we noted 600 cells input and 900 cells input for duplicates and triplicates, respectively). However, as Reviewer #3 points out, it leads to inconsistent input numbers between experiments and made the relevant corrections. Please refer to Figures 1a, 2a and 2b for data with consistent input numbers. Additionally, for figures 1a and 2a, we have decided to report the raw counts of the colonies instead of reporting as ‘% colonies formed relative to Mock’ for improved transparency.

We have clarified on the figure legends for Figure 2 in regards to specifying whether experiments are biological or technical replicates.

With respect to observing different HDR frequencies in between experiments (particularly lower HDR% at low MOI + i53 in Figure 2e), we unfortunately believe that it comes from HSPC donor to donor variability. While cells from most donors behave similarly in their gene-editing outcomes *in vitro*, we sometimes encounter cells from some donors that are more refractory to HDR than others (as observed in Figure 2e). However, despite the variability, we’d like to emphasize that the effect of i53 peptide on improving HDR is consistently observed in all experiments shown in the manuscript including the experiments shown in Figure 2e.

2. In figure 3a, the authors should annotate which are CB and mPB samples. Is there any difference about the source? The addition of i53 increase %HDR of 30-40%, but the basal level of HDR looks gene-specific. The authors should discuss the explanation for this finding. As well as different results in different cell types.

For most of our *in vitro* experiments (unless otherwise noted as in Figure 2d), we used umbilical cord-derived (UCB) CD34+ HSPCs obtained through the Binns Program at Lucile Packard Children’s Hospital at Stanford (PMID 33075720). UCB is a vital source of HSPCs and a favorable graft source in hematopoietic stem cell transplantation. For all transplantation experiments, bone marrow-derived mobilized peripheral blood (mPB) CD34+ HSPCs were used. In the revised manuscript we have tried to make this clearer.

3. In figure 5 results showed no detrimental effect of i53 treatment on HSPC *in vivo* engraftment, with modest improvement (fig.5c) and even lower % editing at low MOI (fig5e). The title of the results section is not supported by data since no improved engraftment of edited cells is biologically meaningful. The experiments were performed by transplanting CB-CD34+ cells. The authors should repeat the experiments with a clinically relevant source, such as mPB.

We apologize for the confusion and mislabeling the figure legends in Figure 5 as cord-blood derived HSPCs. As we originally wrote in our manuscript, the cells that were used for transplantation experiments are all bone marrow-derived mobilized peripheral blood (mPB) CD34+ HSPCs from four independent donors. However, cord-blood derived CD34+ HSPCs were used for all other *in vitro* experiments. We have corrected the legends in Figure 5. We switched to using mPB derived cells for engraftment because those would be used in a translational

setting. As we discussed in response to a similar concern from reviewer 2, the interpretation of results from this heterogeneous model is complicated.

4. Discussion line 293: the statement of improved HDR in vivo is not supported by these data. Line 302: ref De Ravin et al. should be cited also here and those data should be discussed. Line 307: the dramatically improved number of CFU by using i53 is not supported by data, since the graph in figure 2b lacks the control of low MOI-i53. Line 317: the claim of improved engraftment in NSG mice should be corroborated by statistically significant differences and biologically relevant implications.

Line 293: We thank the reviewer for highlighting this point. We have changed the line to say “improve the total number of HDR edited alleles engrafted” which is supported by the data in Figure 5F.

Line 302: Thank you for pointing out this missing this reference Brault et al Frontiers in Immunology (2023) on which we are co-authors. We have included in the text of revised manuscript: “Brault et al (2023) demonstrated that using mRNA to express i53 and mRNA to inhibit p53 directly that they could achieve better genetic engineering frequencies and better engraftment compared to lentiviral modification at the IL2RG locus. In this work, we have identified that i53 peptide can also be used and because of its shorter expression might have a better safety profile than i53 mRNA expression. Future work will test whether combining i53 peptide with p53 inhibition provides a further increase in engraftment of HDR edited cells.” We have cited the work of the Naldini, DeRavin, and Sauvageau labs for their labs work showing the potential benefit of inhibiting p53 (not to be confused with 53BP1).

Line 307: We apologize for the lack of clarity with our claim. In Figure 2b, we reported the values normalized to control HSPCs that have been mock electroporated and treated with i53 peptide. Therefore, we have changed the labels to make it clear in both the figure and its corresponding legend.

Line 317: We have modified the text and conclusions so as to state that there is not statistical significance.

Minor:

Figure 4b: should the authors increase the number of biological replicates, at least n=3?

Thank you for your suggestion. We incorrectly described in Figure 4b that “Data from n=2 biological replicates with two technical replicates”. By biological replicates we were incorrectly referring to biological donors of HSPCs, and technical replicates to the experiments conducted using the same donor. We have therefore made the correction in the figure legend. “Data conducted using HSPCs from two CB donors. For each donor, the experiment was independently conducted two times.”

Methods: supplier for i53 should be indicated.

The i53 peptide was supplied by IDT and their contributions of the peptide as well as their intellectual contributions in analyzing and interpreting the data are reflected in their authorship on the work. We have included in the acknowledgements section that IDT should be contacted if an investigator wishes to test the i53 peptide.

References: accurate check is required, some references are incomplete (#28, 41).

Thank you for pointing out the error. We have corrected the incomplete references.

REVIEWER COMMENTS

Reviewer #1 (Remarks to the Author):

All my concerns have been addressed.

Reviewer #2 (Remarks to the Author):

The authors have provided satisfactory responses to address most of the points raised. However, given the emphasis on therapeutically relevant loci in HSPCs and the application for clinical use (abstract, introduction, discussion...), a more thorough description of genome instability risks should be included given the rather substantial literature on this topic. The risks of genome rearrangement following Cas9/AAV genome editing should be fully disclosed in the introduction since any impact with i53 affects the DNE repair process and safety thereafter. The response provided in point 2 is inadequate.

The assay for detection of translocations (Figure S3c, d) is not relevant here because sgRNAs targeting two different loci simultaneously do not reflect any potential clinical application and is unrevealing for events that may occur as a result of nuclease-induced double-strand breaks. In addition, the authors also referred to AAV insertions that may occur at low frequencies. This information should be provided by sequencing of the target site(s) to determine if i53 affects chromosomal rearrangement at the Cas9 cut site.

The authors conclude that using lower MOI of AAV6 might further decrease random integrations and improve safety. However, your data shows that despite increased p21, lower CFUs with higher MOI (5000), your long term %editing x %engraftment shows the best outcome with the highest MOI (5000) (Fig. 5f). This implies that despite the advantages of lower AAV MOI, a higher MOI (with i53) might still provide the best long term sum-total outcome.

Reviewer #3 (Remarks to the Author):

The authors addressed most of the comments. The revised version of the paper has been improved.

Response to Reviewer Comments

Dear Reviewers,

We greatly appreciate the continued thorough review of our manuscript entitled “Transient Inhibition of 53BP1 Increases the Frequency of Targeted Integration in Human Hematopoietic Stem and Progenitor Cells” for potential publication in Nature Communications. We are pleased that there are just a few extra suggestions. In the revised manuscript, we address your valuable comments through both textual enhancements and the inclusion of experimental data, particularly regarding the safety considerations associated with employing the i53 peptide for transiently inhibiting 53BP1. Below, we present a detailed point-by-point response to the remaining comments.

Reviewer #2 (Remarks to the Author):

The authors have provided satisfactory responses to address most of the points raised. However, given the emphasis on therapeutically relevant loci in HSPCs and the application for clinical use (abstract, introduction, discussion...), a more thorough description of genome instability risks should be included given the rather substantial literature on this topic. The risks of genome rearrangement following Cas9/AAV genome editing should be fully disclosed in the introduction since any impact with i53 affects the DNE repair process and safety thereafter. The response provided in point 2 is inadequate.

Thank you for pointing this out and we sincerely apologize for not adequately addressing these comments in our previous revision. As a response, we have incorporated in the introduction, which states: “53BP1 is a key player in the NHEJ pathway and altering its function could potentially increase the risk of genome rearrangements. Therefore, it is essential to evaluate potential risks and understand the mechanisms that underlie genome instability. Several important genome instability risks in gene editing include: 1) off-target effects, where Cas9 enzyme can unintentionally introduce DSBs at sites that resemble the target sequence, 2) unintended insertions, deletions and rearrangements can result from the error-prone nature of NHEJ repair pathway, and finally 3) unwanted rearrangements and insertions can result from HDR pathway, which might not always be precise.”

The assay for detection of translocations (Figure S3c, d) is not relevant here because sgRNAs targeting two different loci simultaneously do not reflect any potential clinical application and is unrevealing for events that may occur as a result of nuclease-induced double-strand breaks. In addition, the authors also referred to AAV insertions that may occur at low frequencies. This information should be provided by sequencing of the target site(s) to determine if i53 affects chromosomal rearrangement at the Cas9 cut site.

Once again, we express our gratitude to the reviewer for requesting additional data that address the safety considerations related to the use of the i53 peptide. In terms of the relevance of targeting two loci simultaneously (multiplexing), we note that there are clinical protocols in which multiplexing is being tested using nuclease-based editing (including CAR-T trials by Allogene, Collectis, CRISPR, Sana and others). Thus, there is clinical relevance. Complementing the prior existing data, we have now incorporated CAST-seq, a state-of-the-art sensitive sequencing-based assay designed to identify and quantify any chromosomal abnormalities, including instances of off-target and translocation events, that might arise in cells subjected to

i53 peptide treatment during gene editing after creation of a break at a single target site.

Through this assay, we can categorize the clusters of aberrations into distinct group: those resulting from on-target processes, those arising from off-target mediated translocation events, and those attributed to homology-mediated translocation. Due to the assay's high sensitivity, other observed aberrations are classified as non-classified break sites. For a comprehensive understanding of this data, please refer to figures 4e, 4f, and supplementary figures S3e and S3f but in summary, just as for gross chromosomal translocations induced by breaks at two loci, the inclusion of i53 peptide did not increase the gross chromosomal rearrangements at the on-target site when a single break is made.

The authors conclude that using lower MOI of AAV6 might further decrease random integrations and improve safety. However, your data shows that despite increased p21, lower CFUs with higher MOI (5000), your long term %editing x %engraftment shows the best outcome with the highest MOI (5000) (Fig. 5f). This implies that despite the advantages of lower AAV MOI, a higher MOI (with i53) might still provide the best long term sum-total outcome.

We thank the reviewer for the careful reading and analysis of the data. We agree that the engraftment experiments show the effect of i53 and the ability to lower the MOI of AAV6 on engraftment needs continued study. A real challenge of these xenograft studies is the heterogeneity even within a single experiment using a single source of CD34+ cells, not to mention the variability between experiments using different donors. We presented the full set of data for transparency for what we have done and we anticipate that further study will be needed to determine the optimal combination of factors to result in the highest level of engraftment of HDR edited cells. It is likely that the inclusion of i53 in order to lower the AAV6 MOI will only be one part of the solution.

REVIEWERS' COMMENTS

Reviewer #2 (Remarks to the Author):

The authors have adequately addressed the concerns.

Response to Reviewer Comments

We are pleased that we have satisfied the reviewers questions and concerns.